# Invasive earthworms unlock arctic plant nitrogen limitation

Gesche Blume-Werry [1,2], Eveline J. Krab [2,3], Johan Olofsson[2], Maja K. Sundqvist[2,4], Maria Väisänen[5,6] & Jonatan Klaminder [2✉]

Arctic plant growth is predominantly nitrogen (N) limited. This limitation is generally attributed to slow soil microbial processes due to low temperatures. Here, we show that arctic plant-soil N cycling is also substantially constrained by the lack of larger detritivores (earthworms) able to mineralize and physically translocate litter and soil organic matter. These new functions provided by earthworms increased shrub and grass N concentration in our common garden experiment. Earthworm activity also increased either the height or number of floral shoots, while enhancing fine root production and vegetation greenness in heath and meadow communities to a level that exceeded the inherent differences between these two common arctic plant communities. Moreover, these worming effects on plant N and greening exceeded reported effects of warming, herbivory and nutrient addition, suggesting that human spreading of earthworms may lead to substantial changes in the structure and function of arctic ecosystems.

[1] Experimental Plant Ecology, Institute of Botany and Landscape Ecology, University of Greifswald, Soldmannstraße 15, 17487 Greifswald, Germany. [2] Climate Impacts Research Centre, Department of Ecology and Environmental Science, Umeå University, 981 07 Abisko, Sweden. [3] Swedish University of Agricultural Sciences, Department of Soil and Environment, 750 07 Uppsala, Sweden. [4] Department of Earth Sciences, University of Gothenburg, 413 20 Gothenburg, Sweden. [5] Arctic Centre, University of Lapland, P. O. Box 122, 96101 Rovaniemi, Finland. [6] Ecology and Genetics Research Unit, University of Oulu, P. O. Box 3000, 90014 Oulu, Finland. ✉email: jonatan.klaminder@umu.se

N itrogen (N) cycling in arctic soils plays an important role for the global climate system[1] as vegetation changes in this biome directly affect the radiative forcing of our planet[2,3] and the carbon pool in arctic soils plays a profound role in the terrestrial carbon-climate feedback system[4]. Arctic plants and microbes are heavily controlled by N availability[5–9]; hence, disruptions in N cycling can have large consequences for the functioning of tundra ecosystems. Commonly, the low nutrient availability in arctic soils is attributed to the cold climate hampering microbial turnover of nutrient pools[10]. In line with this view, abiotic drivers such as warming[11] or permafrost thaw co-occurring with warming[12] have been shown to increase intrinsic N cycling and N availability for tundra plants. Other studies have explored effects of other abiotic drivers, such as increased external inorganic N inputs[13,14]. Even though biotic drivers of change (e.g., herbivory) have also been acknowledged, no study has so far experimentally quantified potential impacts of invasive soil macrofauna on the intrinsic arctic N cycle—organisms that may provide functions that are currently absent in the decomposition process of tundra soils.

A potentially invasive group of soil macrofauna are earthworms that utilize deeper mineral soil layers (endogeic and anecic species) and thereby transform and restructure nutrient pools in upper soil horizons (henceforth: geoengineering earthworms). Earthworms represent textbook examples of organisms that stimulate N mineralization[15], and can cause increased nitrification[16] or denitrification due to their gut-associated bacterial community[17]. By releasing N from organic compounds earthworms stimulate microbial processes[18] and plant growth[19]. Due to their capacity to both transform and restructure N pools, invasion of geoengineering earthworms into pristine tundra, where they are currently absent, may have substantial effects on arctic soils. Past glaciations eradicated geoengineering earthworms from arctic soils and their slow natural dispersal rates, 5–20 m yr$^{-1}$[20,21], could leave high latitude landscapes without earthworms even over a Holocene time-scale—if not assisted by human mediated dispersal. Despite slow natural dispersal rates, human mediated introduction of earthworms into formerly glaciated landscapes may lead to near complete colonization over a decadal to centennial time-scale[21,22]. Observations also show that geoengineering earthworms, when introduced by humans, have established in arctic soils encompassing North America, Greenland, Iceland, Fennoscandia, and Russia (Fig. 1). Further, global warming is expected to facilitate survival and establishment of southern invaders, such as earthworms, both because of predicted improved abiotic conditions and due to the increasing number of human-mediated dispersal vectors that accompany intensified land-use in the Arctic[23]. Increasing soil temperatures and predicted formation of year-round unfrozen soil layers (talik) in the near future[24] offer examples of potential niches opening in areas where frozen soils have previously limited earthworm establishment. In addition, recent models assessing both soil properties as well as climatic conditions predict that large areas in the Arctic are already suitable for several earthworm species[25]. In other ecosystems, some plants can be negatively affected by earthworm activities[26], which makes the impact of geoengineering earthworms in tundra plant–soil systems largely speculative given the current lack of empirical studies.

Here, we study two species that represent geoengineering earthworms (*Lumbricus* sp. and *Aporrectodea* sp.) which, once introduced by humans, can establish in the Arctic (Fig. 1) and reshape the arctic soil morphology in our study region in northern Sweden[27]. We use an arctic common garden experiment conducted over two growing seasons (Supplementary Fig. 1, Supplementary Table 1) consisting of two vegetation types (heath and meadow) supplemented with $^{15}$N labeled litter. In this setting, we applied geoengineering earthworms to assess their effect on N cycling in dwarf shrub- and forb dominated tundra communities. We hypothesize that, by providing currently missing functions in the decomposition process, these earthworms increase plant N uptake. In addition, we hypothesize that positive effects on plant N uptake are sufficient to enhance plant growth.

## Results

**Earthworm effects on plant N uptake.** Plant N concentration and N sources of the five studied plant species changed already in the first growing season with geoengineering earthworm presence. Without earthworms present, N concentration of the grass *Festuca ovina*, the only species common in both plant communities, did not differ between heath (dwarf shrub dominated) and meadow (forb dominated). However, with earthworms present N concentration in *F. ovina* increased on average by 50% ($F_{1,6} = 43.3$, $P < 0.001$, Fig. 2), more strongly so in the heath than in the meadow (vegetation × earthworm $F_{1,24} = 5.6$, $P = 0.025$; Fig. 2). Both the earthworm additions ($F_{1,6} = 20.9$, $P = 0.004$) and isotopic labeling of the litter layer ($F_{1,24} = 57.1$, $P < 0.001$) induced a significantly higher $\delta^{15}$N in *F. ovina* (Fig. 2). In addition, there was a significant interactive effect of earthworm and litter labeling treatment on $\delta^{15}$N in *F. ovina* ($F_{1,30} = 4.7$, $P = 0.039$), showing that more N from the litter pool was used by *F. ovina* when earthworms were present.

The analyses of four additional plant species that were specific to one of the two vegetation types further indicated that earthworms increase the N concentrations in plants. Of the species only present in the heath vegetation, in addition to *F. ovina*, *Vaccinium vitis-idaea* had higher N concentrations in the presence of earthworms ($F_{1,6} = 13.7$, $P = 0.010$) and while *Vaccinium myrtillus* responded in a similar way as *V. vitis-idaea* and *F. ovina*, the difference was not statistically significant ($F_{1,6} = 5.6$, $P = 0.055$) (Fig. 3). Similar to the effects on *F. ovina* reported above, the increased plant N in the habitat-specific species was also derived at least partly from the litter layer; the $\delta^{15}$N of *Saussurea alpina* did not differ between labeled and unlabeled litter additions without earthworms, but it increased with earthworms (labeling × earthworm, $F_{1,14} = 4.7$, $P = 0.048$) (Fig. 3). No other of the plants specific to the two vegetation types showed this interaction effect (Supplementary Table 4).

There was no earthworm effect on the gravimetric (measured only during the first growing season) or volumetric soil moisture (first season). However, the volumetric soil water content (measured continuously with loggers) suggested a difference between the vegetation types ($F_{1,9} = 16.4$, $P < 0.01$) and the seasonal average water content was higher in the meadow (0.23 m$^3$/m$^3$) than in the heath (0.02 m$^3$/m$^3$). Analyses of litter cover, soil nutrients—using plant root simulator (PRS) probes—and microbial community—using phospholipid fatty acid (PLFA) analysis—after the first season provided additional insights about the processes driving plant N uptake in the presence of earthworms (Table 1). Earthworms reduced the surface litter cover, yet only in the meadow (vegetation × earthworm; $F_{1,38} = 7.9$; $P = 0.008$), but we detected a vertical mixing of surface litter into the rooting zone in the form of belowground earthworm tunnels and castings in both vegetation types (Supplementary Fig. 2). The PRS probes revealed no significant differences caused by earthworms in the availability of soluble inorganic N fractions ($NO_3$ and $NH_4$) and other key macronutrients such as phosphorous, base cations, iron and manganese (Supplementary Fig. 2). Nevertheless, in a complementing leaching study (see Methods) analyses of water soluble N forms showed that N

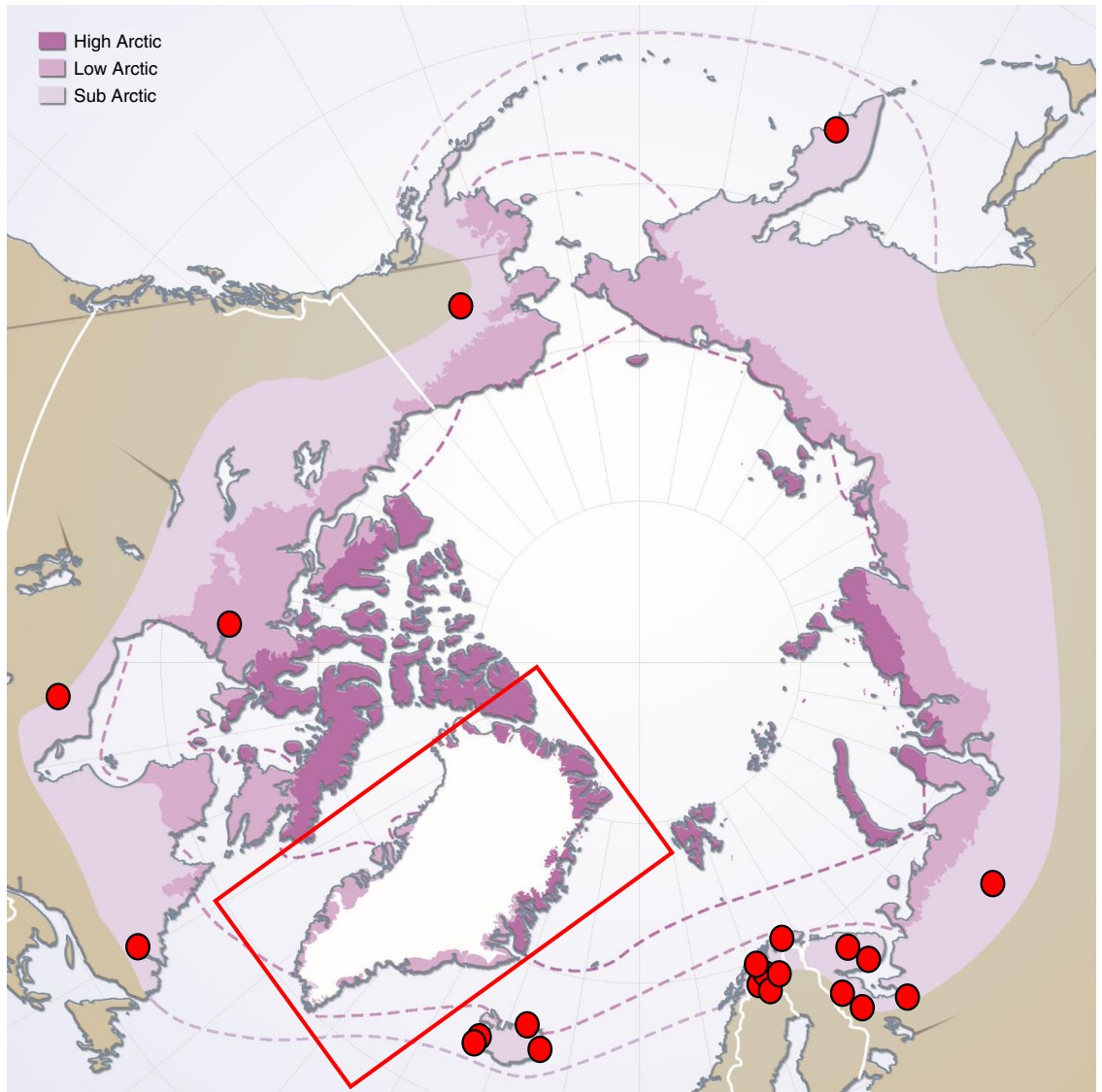

**Fig. 1 Documented presence of geoengineering earthworms in the Arctic.** Sites (red circles) in which *Lumbricus* sp. and *Aporrectodea* sp. have been found (utilizing the mineral soil as a habitat) in the arctic biome (purple shading) divided into three major geographical zones (sub-, low-, and high-arctic). Sites are compiled from previously published studies[27,47,49,52-55] and findings presented in the Global Biodiversity Information Facility database. Red rectangle indicates a reported finding of *L. rubellus* in Greenland where the specific site was not reported[56]. The figure does not intend to provide a complete overview of known populations of geoengineering earthworms, but to illustrate that they can survive in the Arctic (mainly the sub-arctic zone). A common denominator is that earthworms occur adjacent to human introduction points. Thus, the map illustrates that human mediated introductions occur at circumpolar scale. The underlying map showing geographic areas of the Arctic is derived from the Arctic Biodiversity Assessment (http://grida.no/resources/6264).

mineralization was profound in the earthworm casts. Fresh casts (<7 days old) were dominated by inorganic N and had an $NH_4$ concentration >10,000% higher than "earthworm free" litter and humus that were dominated by organic N forms (Supplementary Fig. 3). Nitrate concentrations were minute in comparison to $NH_4$ in casts, litter, and humus. The PLFA analysis from the samples collected from the upper 10 cm horizon (including litter and soil), suggested that earthworms reduced bacterial biomass in the meadow, but not in the heath (vegetation × earthworm; actinomycetes: $F_{1,38} = 4.6$, $P = 0.039$; other bacteria: $F_{1,38} = 4.8$, $P = 0.035$), and had no effect on fungal biomass in either of the habitats ($F_{1,38} = 0.8$, $P = 0.358$). The biomass of actinomycetes, other soil bacteria and fungi was higher in the meadow than in the heath (Table 1, actinomycetes: $F_{1,38} = 38.7$, $P < 0.001$; other bacteria: $F_{1,38} = 26.6$, $P < 0.001$; fungi: $F_{1,30} = 10.4$, $P = 0.002$).

**Earthworm effects on above and belowground plant production.** Within the first season after earthworm introduction, the normalized difference vegetation index (NDVI, a measure of greenness) was significantly higher when earthworms were present ($F_{1,6} = 6.4$, $P = 0.045$; Table 1), while abundances of graminoids, forbs, deciduous dwarf shrubs, evergreen dwarf shrubs, mosses, and lichens did not change in response to earthworms (Supplementary Table 3). After the second season with earthworms, the maximum height of floral shoots of *Deschampsia flexuosa* was about three times higher when earthworms were present in the meadow (vegetation × earthworm; $F_{1,38} = 9.3$, $P = 0.004$) while the number of floral shoots remained similar between treatments (Fig. 4a, b). In contrast, the number of floral shoots of *F. ovina* was greater in the presence of earthworms ($F_{1,6} = 4.7$, $P = 0.043$). As would be expected if plant

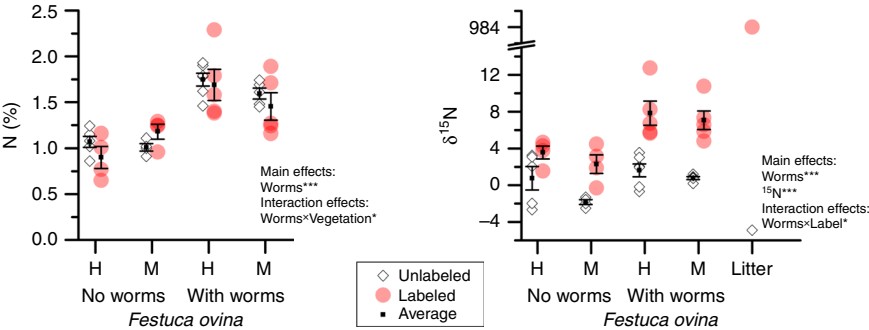

**Fig. 2 Earthworm effects on nitrogen uptake of *Festuca ovina*.** Shown are earthworm effects on N content (%) and $\delta^{15}$N of *Festuca ovina*, a plant present in both of the studied vegetation types, i.e., heath (H) and meadow (M). Individual replicates of samples from unlabeled (white diamond shape) and $\delta^{15}$N labeled (red circle, where darker values indicate overlapping data) mesocosms are shown with average values (black square, ±std. err). Main effects and interaction effects are presented as text for each panel (effect of the labeled litter is indicated using the $^{15}$N symbol) along with symbols indicating significance level ($P < 0.05$ are shown using *, and significance levels <0.001 are indicated as ***). Note the cut off in the *y*-axis to show the $\delta^{15}$N signatures of the unlabeled and the labeled litter. Source data are provided as a Source Data file.

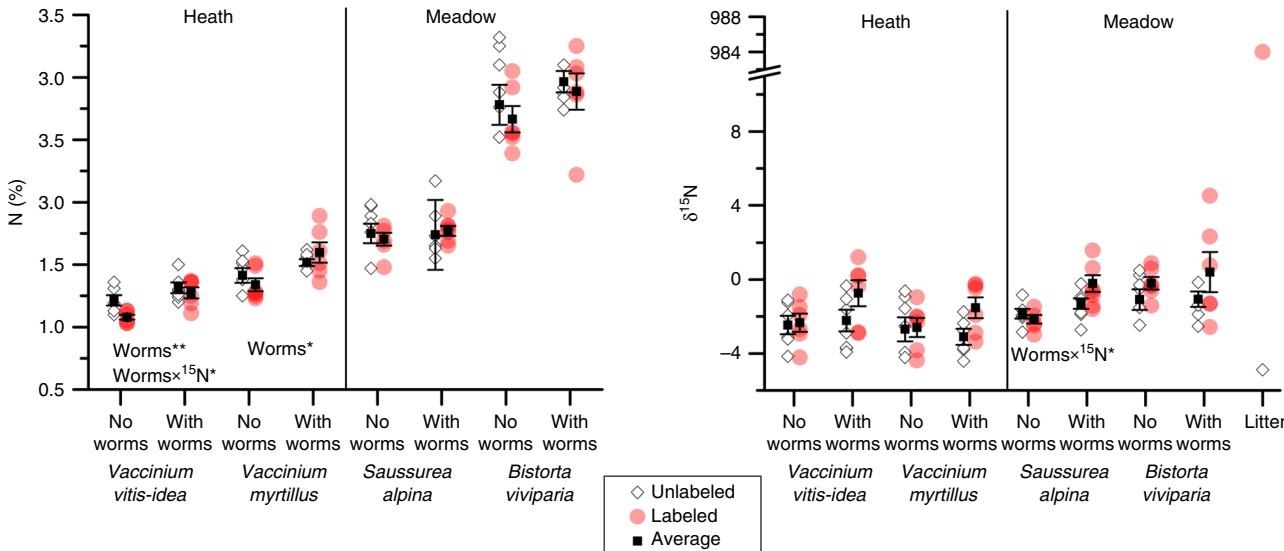

**Fig. 3 Earthworm effects on nitrogen uptake of common heath and meadow species.** Nitrogen content and $\delta^{15}$N signatures of dwarf shrubs found only in heath (*V. vitis-idea* and *V. myrtillus*) and forbs found only in meadow (*S. alpina* and *B. vivipara*) in response to geoengineering earthworms. Note that these plant species were only found in one of the two vegetation types and thus did not allow comparison of effects between vegetation types. Individual replicates of samples from unlabeled (white diamond shape) and $\delta^{15}$N labeled (red circle, where darker values indicate overlapping data) mesocosms are shown with average values (black square, ±std. err). Main effects and interaction effects are presented as text for each panel (effect of the labeled litter is indicated using the $^{15}$N symbol) along with symbols indicating significance level ($P < 0.05$ are shown using *, and significance levels <0.001 are indicated as ***). Note the cut off in the *y*-axis to show the $\delta^{15}$N signatures of the unlabeled and the labeled litter. Source data are provided as a Source Data file.

**Table 1 Effects of geoengineering earthworms on litter cover and microbial and plant communities.**

| Significant factors | Litter cover (%) V, E, V × E | | Actinomycetes (nmol/g soil) V, V × E | | Bacteria (nmol/g soil) V, V × E | | Fungi (nmol/g soil) V | | NDVI (index) E | |
|---|---|---|---|---|---|---|---|---|---|---|
| Heath | 3.6 | ±0.7 | 2.5 | ±0.2 | 45 | ±2 | 11 | ±1.1 | 0.79 | ±0.01 |
| Heath + E | 2.9 | ±0.6 | 2.8 | ±0.3 | 52 | ±7 | 16 | ±1.9 | 0.82 | ±0.01 |
| Meadow | 9.8 | ±0.2 | 6.2 | ±0.7 | 90 | ±9 | 23 | ±3.2 | 0.77 | ±0.01 |
| Meadow + E | 6.3 | ±0.7 | 4.5 | ±0.4 | 68 | ±6 | 21 | ±4.2 | 0.81 | ±0.01 |

Shown are effects of vegetation type (V) and earthworm treatment (E) at a community level on litter, microorganisms and the normalized difference vegetation index (NDVI). Litter coverage is expressed as a proportion of the mesocosm soil surface. Abundances of actinomycetes, bacteria, and fungi are measured using phospholipid fatty acid analysis (PLFA), while the photosynthesis activity (greenness) of the plant community was measured using the NDVI, which is based on the plants' ability to reflect near-infrared light and adsorb red light. Values are mean ± standard error. Statistically significant factors ($P < 0.05$) in the linear mixed effect model are given in the row above the treatments.

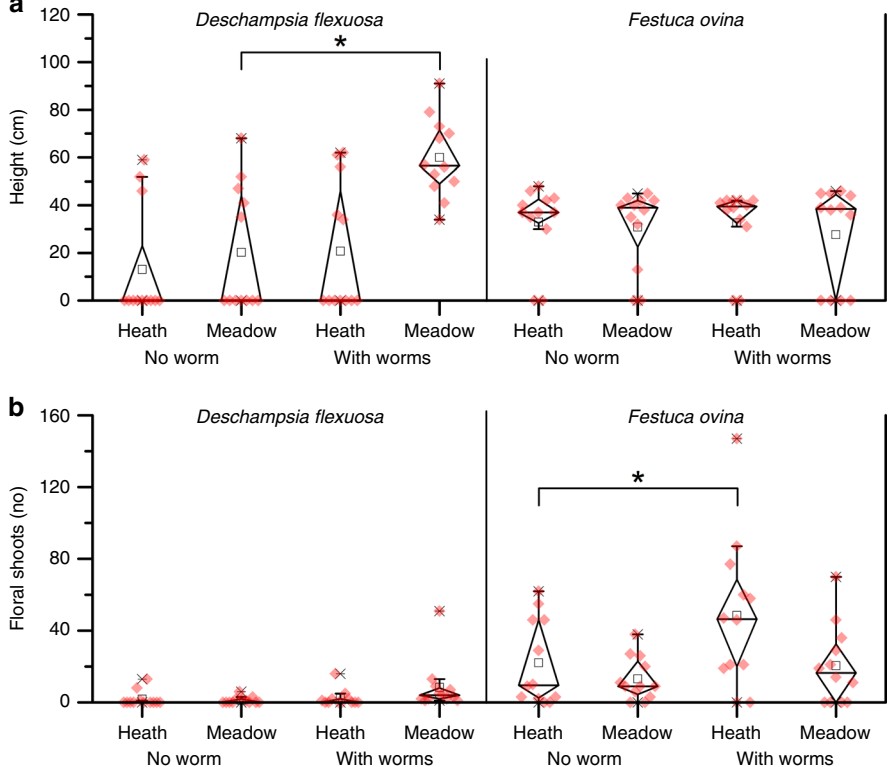

**Fig. 4 Aboveground plant growth responses to earthworm additions.** Boxplot showing **a** the maximum height and **b** the highest number of floral shoots (lower panel) of graminoids (*D. flexuosa* and *F. ovina*) per mesocosm. Diamond box indicate the 25%- to 75%-percentiles (whiskers show 99% percentile) and black rectangle the median value. Mesocosms lacking the studied plants are given a value of 0. Significant effects are indicated with *($P < 0.05$). Source data are provided as a Source Data file.

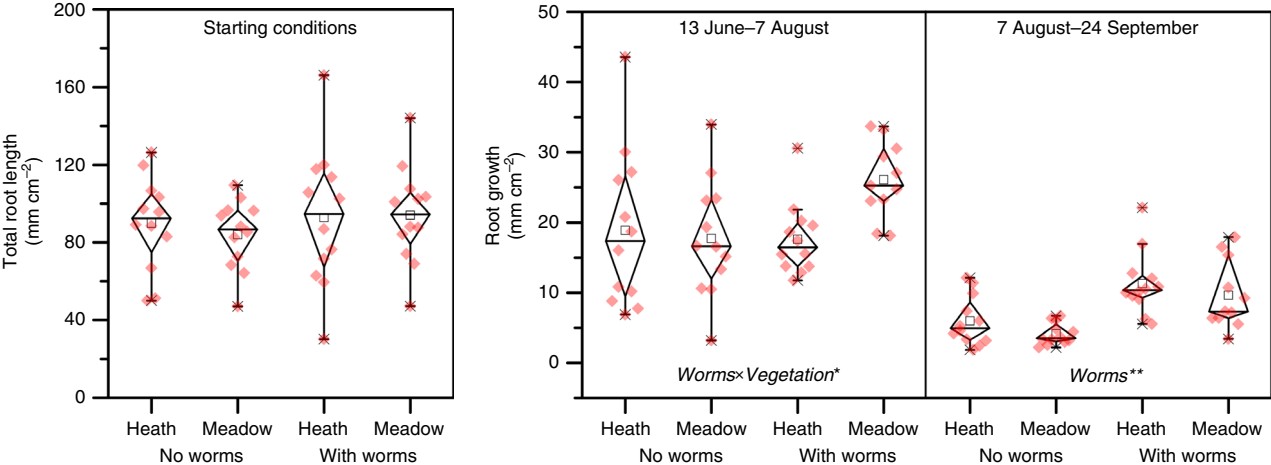

**Fig. 5 Belowground plant growth response to earthworm additions.** Boxplot showing the fine root length at the onset of the experiment before the earthworm addition, and the fine root growth after earthworm addition in heath and meadow vegetation (mesocosm with labeled and unlabeled litter are pooled together). Periods when growth was measured are shown at the top. Diamond box indicate the 25%- and 75%-percentile (whiskers indicate the 99% percentile) and black rectangle show the median value. Also shown are all individual measurements (red diamonds). Treatment effects are written at the base of the panels and the level of significance indicated with *($P < 0.05$) and **($P < 0.01$). Source data are provided as a Source Data file.

N availability was driving the observed growth responses, the N-content of *F. ovina* correlated positively with the number of floral shoots ($R^2 = 0.518$, $P = 0.001$). In addition, the N content of *F. ovina* correlated positively with the NDVI of the studied plant communities ($R^2 = 0.487$, $P = 0.002$).

At the onset of the experiment (13 June 2017, Supplementary Table 1), total fine root length did not differ between the vegetation types or between the earthworm treatments (Fig. 5). After 55 days, fine root growth was higher in meadow including earthworms than in meadow without earthworms (vegetation × earthworm; $F_{1,6} = 4.5$, $P = 0.042$). After 103 days, fine root growth in both vegetation types was almost twice as high under earthworm presence than without earthworms ($F_{1,6} = 8.3$, $P = 0.028$).

## Discussion

Our results suggest that geoengineering earthworms and their activity increase plant N uptake from tundra soils, as earthworm presence increased plant N content, altered the $\delta^{15}N$ of plant leaves, and boosted fine root growth already within the first season. The increased uptake of N was associated with increased individual plant height and amount of floral shoots and this enhanced performance of individual plants was also accompanied by community level increased NDVI (a measure of vegetation greenness). The most likely driver behind these effects is a boosting of N mineralization processes by earthworms, a process identified both by previous laboratory experiments[28] as well as our comparison of $NH_4$ in earthworm casts, litter and humus (Supplementary Fig. 4).

Geoengineering earthworms doubled leaf tissue N concentrations in *F. ovina*—an effect larger than the difference seen in this species caused by the growth conditions in the two studied vegetation types—and, thus appear at least as important as the dwarf shrub- and forb-driven effects currently dictating N cycling in tundra ecosystems[29,30]. Given the strong N limitation in arctic systems[5,7,9] the increased plant N uptake triggered by earthworms is, indeed, expected to increase above- and belowground plant growth (hypothesis 2). Yet, the near doubling in fine root growth in fall and the more than doubling in length of *D. flexuosa* and number of shoots of *F. ovina* that we observed are surprisingly strong compared to the inherent differences between the studied dwarf shrub and forb vegetation communities. The effects we observed on root growth and above ground biomass were also strong compared to North American forests, where earthworms can even have negative effects on fine root growth[31], and to agricultural systems where earthworms cause on average 23% increase in aboveground biomass[19]. Similar to manipulations of abiotic factors such as nutrients and temperature[32] some of the effects of earthworms on plants seem species-specific. Some species increased N uptake in the presence of earthworms (*F. ovina*, *V. vitis-idaea*), while others (*S. alpina*) only changed their N sources as indicated by the changes in $\delta^{15}N$. Similarly, growth responses to earthworms also seem species specific since *D. flexuosa* increased in length of floral shoots, while *F. ovina* produced more of them. This asymmetric response between species suggests that if these plant responses are permanent, tundra plant communities could change in the long-term, possibly increasing the abundances of graminoids as seen in earthworm invaded North American forests[26].

The effects of earthworms on fine root production in dwarf shrub and forb communities were notably stronger than the effects seen in local warming experiments, simulating autumn warming[33] or variation in climatic conditions prevailing along a 500 m elevation gradient[34]. To further illustrate the importance of geoengineering earthworms in comparison to other environmental factors/drivers in our study area, we extracted plant community N (a weighted average N content calculated using coverage of plant functional groups as weights) and NDVI data (vegetation greenness) from previous local studies. These were situated in comparable vegetation settings within the same region (<20 km from our experiment) reducing any bias caused by context-dependent responses (Supplementary Tables 3 and 5). This comparison revealed that the effects on N content and NDVI caused by geoengineering earthworms were as strong, or even stronger, than other important environmental drivers in the Arctic, including warming, herbivory and fertilization (Fig. 6a, b). For example, earthworms increased plant community N concentrations over three times as much as a 3 °C increase in summer temperatures[35,36] or extensive fertilization with reindeer feces[37]. Moreover, the increases in NDVI caused by the earthworms were similar to the differences in greenness caused by a

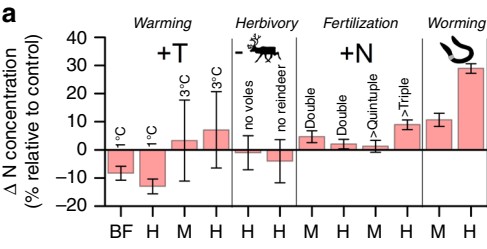

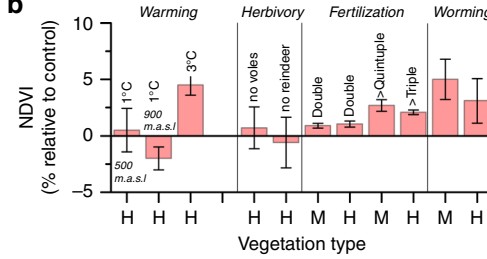

**Fig. 6 Effects of earthworms compared to other environmental drivers in a changing Arctic.** Shown are a conceptual comparison of effects (percentage change relative to average of control) caused by earthworms (worming) with effects seen in previous local experiments in sub-arctic northern Sweden simulating arctic environmental change including warming, herbivory, and fertilization. **a** Effects (mean ± std. err) on plant community N expressed as a function of vegetation type (BF = sub-arctic birch forest, H = heath, and M = meadow). Nitrogen data for warming are from open top chambers[35] and from an alpine altitudinal gradient[36], the herbivory experiment was conducted using fences excluding voles and reindeer (shown effects are illustrating when grazers are present in relation to the ungrazed control)[35] and reindeer feces were used in the fertilization experiment, where additions corresponded to about double and four times natural abundance[37]. **b** Effects (mean ± std. err) on NDVI (i.e., greenness) expressed as a function of vegetation type. Experiments are similar as above with the addition of data from an open top chamber experiment simulating a mean annual temperature increase of 1 °C conducted at both an altitude of 500 and 900 m.a.s.l. Source data are provided in Supplementary Table 3, Supplementary Methods 1 and as a Source Data file.

3 °C warming along natural elevation gradients and more substantial than effects of herbivory and fertilization (Fig. 6b). Even expanding our comparison from local experiments to large-scale datasets further suggests that the worming effect on NDVI and community N is substantial compared to the effects of warming. Plant community N increased by about 30% in the presence of earthworms in our study, which is larger than the average 15% increase in response to summer warming from mean temperatures of 3–10 °C occurring across the arctic biome[38]. The earthworms also caused an increase in NDVI of 0.02–0.04 units (corresponding to 3–5% change) in our study, which is comparable to the greening accompanying shrub expansion and increased plant growth during the last decades in the Arctic (typically between 0.17 and 0.84% yr$^{-1}$ increase in NDVI)[39,40].

Earthworms boost the plant N uptake through two apparent processes in our study. Firstly, by stimulating mineralization of litter and humus into $NH_4$-enriched casts; and secondly by vertically translocating nutrients from aboveground litter and fertile casts into the rooting zone so that they become more accessible for plants. Firstly, earthworms boost mineralization strongly in arctic soils as verified by their casts that had $NH_4$ concentration >10,000% higher than the bulk soil (Supplementary Fig. 3)— importantly, this enrichment is one magnitude higher than observed in agricultural ecosystems[41]. Secondly, earthworms facilitate plant use of N from aboveground plant litter as revealed

by decreased litter cover and a change in plant–$\delta^{15}$N toward the signature of the labeled litter. Indeed, an increased importance of litter derived N is expected from geoengineering earthworms considering their ability to mineralize litter, but also annual mixing ~0.6 cm of the surface soil into deeper soil layers[42]. We confirmed this mixing to be an active process in our tundra experiment by observing reduced litter cover in the meadow (forb dominated) in parallel with deposition of earthworm casting directly within the rooting zone (Supplementary Fig. 2). Because the increases in plant N concentrations do not fully match the increases in $\delta^{15}$N, also other, older soil N pools were mobilized and deposited as casts—especially in the heath (dwarf shrub dominated) where litter cover decreased less evidently. We acknowledge that dead and decomposing earthworms could be an additional N source in the experiment (as also under field conditions), but we do not believe this is a major contribution for two reasons. Firstly, the N taken up by plants was, at least partly, derived from plant litter. Secondly, our belowground imaging revealed living earthworms and formation of earthworm tunnels at the same time as positive effect on plant N uptake occurred, which was during both the peak and late growing season prior to the first winter of our experiment when soil–frost induced mortality is expected.

The mobilization of N from aboveground litter and the high availability of mineral N in earthworm casts (as measured in the leaching study) were associated with a higher plant N content, although we did not record any increases in mineral N in the PRS probes targeting the in situ soil solution. This is likely caused by a rapid plant uptake of the N made available by the earthworms, resulting in little N diffusion through the soil solution to the PRS probes. These results corroborate previous studies showing that N from urine and droppings of reindeer can increase plant N uptake and growth without any measurable effects on soil N availability[43–45]. While plant N concentrations and growth showed strong positive responses to the earthworms in both vegetation types, microbial biomass increased with earthworm presence in the heath and decreased in the meadow. We acknowledge that these observed microbial effects could be linked to both a direct effect of earthworms resulting in different responses among types of microbes, or indirect effects where the microbial community responds to changes in soil organic matter quality and the observed plant community specific effects on fine root growth. Considering the earlier response in fine root growth in the meadow, fiercer plant competition for N during the summer seems as a plausible process contributing to reduced microbial biomass in these plant communities.

The finding of a strong positive effect of geoengineering earthworms on plant N uptake is an important advancement in our understanding of plant N limitation in tundra soils. It shows that plant uptake of N from litter and soil pools can be greatly enhanced by introducing detritivores that are able to accelerate mineralization of organic matter and excrete processed material, casts, into the rooting zone. To synthesize, the absence of these functions represents a strong bottleneck in the plant N cycle in the Arctic. Our findings do not contradict the current view that the N (and carbon) cycle is constrained by temperature dependent microbial processes, but add a new central process—a great boost in the intrinsic N cycle can be achieved by changes in the soil macrofauna alone, without any direct impact of temperature per se. This increased knowledge is important as it implies that any mechanistic model predicting future changes in the arctic N cycle and subsequent effects on greening, plant communities or carbon cycling, needs to consider dispersal scenarios for soil macrofauna adding novel functions into the current decomposition process in tundra soils. By unlocking the physical entrapment of nutrients, occurring both at a vertical scale (above the

rooting zone) and within soil organic matter that can be mineralized inside the earthworm gut, geoengineering earthworms provide plants access to new nutrient pools. Conceptually, this effect can be compared with processes generated during permafrost thaw, where thawing soil layers release soil nutrient pools previously inaccessible for plant roots which is recognized to have a strong impact on the plant N cycle[12]. However, in contrast to permafrost thaw, where effects are generated at greater soil depths and thus primarily affect deep-rooted plants[46], geoengineering earthworms affect nutrient cycling within the upper few decimeters of soil where most plants have already allocated their roots. That geoengineering earthworms survive in discontinuous permafrost zones of Scandinavia, Russia, and North America when introduced by humans[27,47–49], shows that suitable habitats already exist in the Arctic, and highlight that their lagged northward dispersal needs to be considered when fully understanding long-term nutrient cycling at high latitudes. The demonstrated strong effects on the tundra plant N uptake by geoengineering earthworms show that natural or human-aided dispersal of novel, large detritivores into tundra soils can have substantial impacts on the tundra ecosystem. From human-mediated invasions in other environments, we know that near complete (>80%) colonization of suitable habitats can occur over a decadal to centennial time-scale due to multiple introduction points[22], indicating that earthworm invasions can proceed over a time-scale similar to that of climate change. Importantly, expected impacts from invasive earthworms on the plant–soil N cycle exceed those generated by climate warming and other currently recognized drivers of environmental change in this biome.

## Methods

**Study site.** This study was conducted at the Abisko Scientific Research Station (68° 21′17.0″ N; 18°48′54″ E) about 200 km north of the Arctic Circle. The station is located within the sporadic permafrost zone, with a mean annual temperature around 0 °C and mean annual precipitation of 335 mm (1981–2010, Abisko Scientific Research Station, 380 m.a.s.l.).

**Common garden experiment.** We established a mesocosm ($N = 48$) experiment in the experimental garden of the Abisko Scientific Research Station. An overview of the experimental design is shown in Fig. S1 and a timeline for all activities is shown in Table S1. The mesocosm experiment consisted of natural tundra vegetation–soil monoliths from two vegetation types (meadow and heath) installed during the autumn of 2013. Monoliths for the mesocosms were collected in the Kärkevagge valley, about 20 km northwest of Abisko, Sweden, (at around 68°24′36″ N; 18°19′ 11″ E) at about 700 m elevation. Here, we selected representative patches of both heath and meadow vegetation. The heath was dominated by dwarf shrubs (*Empetrum hermaphroditum*, *V. myrtillus*, *V. vitis-idaea*), with a considerable amount of graminoids (*Carex bigelowii*, *Deschampsia flexuosa*, *Festuca ovina*) and bryophytes (*Pleurozium schreberi*, *Hylocomium splendens*). The meadow was dominated by forbs (*Alchemilla glomerulans*, *Bistorta vivipara*, *S. alpina*) and graminoids (*C. bigelowii*, *D. flexuosa*, *F. ovina*). Soils were pooled by vegetation type and homogenized to ensure a similar depth of soil under each vegetation patch regardless of their original soil depth. Mesocosms were constructed using polypropylene plastic boxes (50 × 39 × 30 cm, with 10 drainage holes with a diameter about 1 cm at the bottom) with a layer of weed cloth, a fine layer (c. 1 cm) of sand, and the mixed heath and meadow soil (intact vegetation including organic sods of 8 cm, placed on top of an Ah mineral soil horizon of 22 cm). The thickness of the O horizon are typical to that of local soils prior to earthworms invasions[27]. Average soil organic matter content for the upper 10 cm for the heath and meadow mesocosm were 12.1 ± 0.8% (mean ± std err) and 11.4 ± 1.3%, respectively.

In the common garden, eight wooden raised beds (300 cm × 160 cm, 40 cm high) were prepared, insulated with 5 cm thick styrofoam and filled with sand. The mesocosms were then installed into the raised beds at a randomly determined location. Either five or seven mesocosms were installed per raised bed and these mesocosms presented both heath and meadow vegetation in each bed at approx. equal numbers. The soil surface of the mesocosm was levelled with the surrounding sandy beds to maintain realistic soil temperature fluctuations. The mesocosms were allowed to recover from disturbance between the summer of 2013 and spring 2017 prior to the onset of the experiment. During the experiment soil temperature (−1 cm) and volumetric soil moisture content (down to −10 cm) were recorded at an hourly interval using data loggers (Em50 ECH2O) equipped with temperature (EC-5) and soil moisture (ECT) sensors (Decagon Systems, Washington state, US). See Supplementary Fig. 1 for location of loggers.

**Earthworm treatments**. Earthworms were first introduced to the mesocosms on 9th of June 2017 at relevant densities. The earthworm species' used (*Aporrectodea* sp. and *Lumbricus* sp.) are observed to radiate out from anthropogenic sources in the surrounding arctic environment with four known invasion sites situated within 12 km from our mesocosm experiment[27]. All earthworms used in this experiment were collected manually in forest adjacent to farmlands within the municipality of Umeå (63°50′17.7″ N; 20°18′46″ E), Sweden. In each mesocosm, we added a total earthworm fresh weight of 24.2 g ± 0.3 (mean ± SE) of endogeic *Aporrectodea* sp. (*Aporrectodea trapezoids*, *Aporrectodea tuberculata*, *Aporrectodea rosea*, 16–17 individuals per mesocosm) and epi-endogeic *Lumbricus rubellus* (27–29 individuals per mesocosm), corresponding to densities of 87 and 140 individuals m$^{-2}$, respectively. As a comparison, measured earthworm densities dominated by *Aporrectodea* sp. frequently exceed 200 individuals m$^{-2}$ in nearby (<12 km) arctic earthworm invasion gradients[27]. One day prior to worm addition, all mesocosms were watered with 10 L each (corresponding to 51 mm precipitation), to prevent desiccation of earthworms directly after release. To prevent earthworms from migrating into mesocosms designated as earthworm-free controls, we assigned whole raised beds as either earthworm addition or controls resulting in four raised beds with and four without earthworms.

In contrast to invasive earthworms populations currently thriving in the local sub-arctic environment[27], the earthworms in the mesocosm were unable to migrate below a soil depth of 30 cm. Given that no behavior based strategies could be adopted to cope with the low winter temperatures (down to −15 °C in the mesocosm) and soil frost in the 0.3 m deep mesocosms (soil frost extended down to 1 m during the first winter according to data from Abisko research station, measured <100 m from the experiment), eradication of the earthworm population during winter was considered unavoidable. Therefore, addition of earthworms was repeated in the following season (2018) to assure presence of living individuals. Here, we added earthworms to each mesocosm, corresponding to a fresh weight of 11.7 ± SE 0.3 g on 16 June 2018, to compensate for winter mortalities.

**Addition of $^{15}$N labeled litter**. To assess eventual impact of plant uptake of N from the litter layer, we introduced $^{15}$N labeled coarsely ground plant material (12 g) into half of the mesocosms on 19 June 2017 to isotopically spike the litter layer, while the remaining mesocosms received similarly ground but unlabeled litter (12 g). The litter added was a mixture of plant litter from meadow and heath vegetation with very similar species composition as the one used in this study[43]. The added litter had an atom % $^{15}$N of 0.36 ± 0.00 (unlabeled control, average ± SE) and 0.73 ± 0.00 (labeled litter), and a $\delta^{15}$N of −4.89 ± 0.31 (unlabeled control) and 984.44 ± 12.11 (labeled litter). Besides the $^{15}$N composition, there was no major difference in the N content between the two litter types (unlabeled: 0.97 ± 10% and labeled 0.88 ± 0.01%).

**Plant–root-simulator probes**. We used PRS$^{TM}$ soil probes (Western Ag Innovations Inc., Saskatoon, Canada), commercially manufactured ion-exchange membranes, to assess availability of inorganic forms of nitrogen (NO$_3$–N and NH$_4$–N, as well as Ca, Mg, K, P, Fe, and Mn). We installed two pairs in each mesocosm which were pooled before extraction. PRS-probes were buried from 16 June 2017 to 3 October 2017 at a depth of 4–10 cm.

**Soil sampling**. We collected soil samples to analyze the effects of earthworms on the microbial community composition on the 18 August 2017. From each mesocosm, a composite soil sample was taken by coring to 10 cm depth (four cores, corer diametre 1 cm), including the litter layer but removing the mineral soil horizon and pooling these samples together per mesocosm. In the laboratory, plant roots and litter were manually removed from each composite sample, a sub-sample was analyzed for soil moisture (105 °C, 18 h) and organic matter content (475 °C, 4 h) and the remaining soil sample was frozen (−20 °C) within four days of sampling and consequently freeze dried for PLFA analyses, conducted at the Biogeochemical Analyses Laboratory at the Swedish University of Agricultural Science, Umeå. PFLA analyses are reported in nmol/g dry soil.

**Plant foliar nutrient content analyses**. We collected green foliage from two to three plant species in each mesocosm at the end of the growing season (21 August 2017). From the meadow mesocosms, we collected the leaves of the two forbs that were present in the highest fraction of the meadow mesocosms; i.e., *S. alpina* and *B. vivipara*. From the heath mesocosms, we collected the leaves and stems of two dwarf shrub species, which were found in the highest fraction of the heath mesocosms, i.e., *V. vitis-idaea* and *V. myrtillus*. From both vegetation types, we collected the leaves of a jointly common graminoid; *F. ovina*. Thereby, our leaf samples at each vegetation type represented plant species constituting on average 60% of the vegetation cover. Since present in both vegetation types, *F. ovina* allowed comparison of plant N acquisition between heath and meadow without the interference caused by interspecific variation. Within each mesocosm, we collected ≥ten leaves of randomly distributed plants with fully expanded green leaves. All collected material was immediately air-dried ≥22 °C. Leaves were sorted from stems (*V. myrtillus* and *V. vitis-idaea*), dried (approximately 48 h, 60 °C) and milled into a fine powder (Bertin precellys 24). The ground materials were subsequently analyzed for total N and $\delta^{15}$N (Isotope ratio mass spectrometer DeltaV, Thermo

Fisher Scientific, Bremen, Germany; and Elemental analyzer Flash EA 2000, Thermo Fisher Scientific, Bremen, Germany).

**Aboveground plant sampling**. In 27 July 2017, a vegetation and litter survey was conducted by the point intercept method[50] in all mesocosms. We used two 50-cm wide rows of ten vertical pins at every 10 cm and counted the total number of times each species was intercepted by 20 pins. The total number of hits was normalized to hits per 100 pins in each mesocosm. This data were further used to calculate the abundance of plant functional groups (graminoids, forbs, evergreen, and deciduous dwarf-shrubs, mosses, and lichens) at the mesocosm level. For each mesocosm, we also measured NDVI (Normalized Difference Vegetation Index) from 0.5 m above each mesocosm using a hand-held pole and two channel sensors (SKR 1800D/SS2, SKL925 logger, SpectroSense2, Sky Instruments, Llandrindon Wells, Wales UK). This index measures the differences between near-infrared light (which vegetation strongly reflects) and red light (which vegetation absorbs), where higher values are indicative of higher photosynthetic activity for the plant community (used as a measure of vegetation greenness).

As an additional proxy for plant growth, we measured the length of the highest individuals and floral shoot numbers of *F. ovina*, and *D. flexuosa* (species that occurred in most of the mesocosms) in late autumn of 2018.

**Below ground plant production**. Total fine root length and fine root growth were measured on 13 June, 7 August, and 25 September 2017 using minirhizotrons (Bartz Technology Corporation, Carpinteria, CA, USA), a nondestructive method for observing temporal development of fine roots. These are transparent tubes with an inner diameter of 5 cm, which were placed horizontally at a depth of 10 cm into the mesocosms during their initial installation.

**Nitrogen in earthworm products**. To assess mineralization and nitrification rates in earthworm casts in comparison to litter and humus we conducted a water leaching experiment. All material was collected in situ from the earthworm invasion gradient at Jiebren, situated <10 km from the mesocosm experiment, described in detail elsewhere[27]. Fresh excreted casts were gathered from collected earthworms (*Aporrectodea* sp.) excreting cast directly into vials, while aged earthworm casts deposited during the season on top of litter was collected manually from the soil surface of forb and dwarf shrub vegetation communities growing under sparsely growing sub-alpine birch forest. Litter and humus (sieved through a 2 mm mesh) from meadow and heath vegetation communities was obtained from uninvaded soil behind a stream functioning as a migration barrier[27]. Sampled material was homogenized and subsectioned. One sub-sample was shaken in Milli Q water (sample v-water v, 1:5) for 2 h, centrifuged at 4000 rpm and the supernatant filtered through a 0.2 μm filter to remove bacteria. Sample was directly frozen after filtering and analyzed for inorganic N using a segmented flow analyzer (QuAAyro39, Seal Analytica) and total N was measured using a Formacs HT-I (Skalar) with a mounted nitrogen detector (ND 25). Organic N was calculated by subtracting the inorganic N from the total N.

**Statistics**. The effect of earthworms, vegetation type and $^{15}$N labeling of litter (labeled and unlabeled) on plant N concentrations, plant $\delta^{15}$N, soil nutrient availability, abundance of bacteria, fungi, litter, and plant functional types and height of graminoids was tested with a linear mixed effect model using the lme command from the nlme package[51] within the statistical environment R (R Core Team 2018). Earthworm treatment, vegetation type, and $^{15}$N labeling was treated as fixed categorical factors and block was treated as a categorical random factor to account for the spatial autocorrelation within blocks. Heteroscedasticity was tested by inspecting residuals and data was log transformed in a few cases to achieve homoscedasticity. The number of floral shoots was tested using the glmmPQL function in the statistical package MASS (Venables & Ripley 2002) within the statistical environment R (R Core Team 2018).

## Data availability

The source data underlying results shown in Figs. 2–6 are provided as a Source Data file, and uploaded on figshare [https://figshare.com/articles/Source_data_Blume-Werry_et_al_Invasive_earthworms_unlock_arctic_plant_nitrogen_limitation_xlsx/11956344].

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

## Acknowledgements

We thank Lapplandstiftelsen, Swedish Research Council (2017-04548), FORMAS (2018-01301 and 2013-00533), Göran Gustafssons Stiftelse för natur och miljö i Lappland, the Gunnar och Ruth Björkmans fond för norrländsk botanisk forskning, and Oscar och Lilly Lammstiftelse for financial support. We thank Dr. Ann Milbau for her input during construction of the mesocosms. Plant drying and grinding was conducted in the laboratory facilities of Finnish Natural Resources Institute, Rovaniemi. We thank Alisa Brandt, Manuel Görich, Anna Ley, Laurenz Teuber, and Veronika Vikuk for practical help, and the Abisko Scientific Research Station for support. Open access funding provided by Umea University.

## Author contributions

G.B.W. designed, initiated, and maintained the experiment. G.B.W., E.K., J.O., M.S., M.V., and J.K. conducted the fieldwork statistical analyses was done by J.O. J.K. wrote the paper in collaboration with G.B.W., E.K., J.O., M.S., and M.V.

## Competing interests

The authors declare no competing interests.
