## [Peer Review File · Nature Communications]

Reviewers' comments:

Reviewer #1 (Remarks to the Author):

General comments

This manuscript by Blume-Werry reported the effects of earthworms on nitrogen cycle in the arctic ecosystems. The authors found that worming effects stimulated plant N concentrations and improved plant production. The finding is interesting, novel and deserved to be published. However, to get this manuscript published in a high-profile journal like Nature Communications, I think it is highly necessary to address the following three major comments.

First, the authors should provide strong evidence that the invasion of earthworms could widely exist in arctic ecosystems, which is the basis for the current study. How frequent will the earthworms be observed in current and future climate scenario? How about the invasion rate of earthworm across the arctic region? How general is the pattern of earthworm invasion? Do the authors have some actual field observations, or model predictions? This issue is crucial for current study, and the authors must show these background data so as to help the readers understand the principle of current study. BTW, to my experience on the Tibetan Plateau, we have very few chances to observe earthworms in this largest alpine permafrost region of the Northern Hemisphere.

Second, the authors should provide more evidence that earthworms could alter N cycling processes in arctic ecosystems, which could help to understand why plant N concentration increased under 'earthworm' treatment. Although the authors showed that worming effects led to the elevated leaf N concentrations in several species, it is still unclear what kind of ecological processes drove this N increase. Particularly, Table S1 revealed that worming effects did not alter soil N availability. To my deduction, the possible reason for the increased leaf N concentration is that earthworms stimulated nutrient release from litter or soil organic matter mineralization/nitrification? However, there is no evidence whether and how worming effects altered these two key processes. I strongly suggest to measure nutrient release rate from litter, and also determine soil nitrogen transformation processes (using N isotope technique) with and without worms treatment. These data are key for understanding the patterns observed in this study.

Third, the authors should provide strong evidence to illustrate the linkage between changes in leaf nutrient and plant growth dynamics under earthworm treatment. Currently, the authors observed the increases in leaf N content and floral shoots of *Festuca ovina*. Is there any association between the increases in floral shoots and leaf N content of *Festuca ovina*? Also, the authors observed the increases in plant height of *Deschampsia fleuosa*, but did not show us whether and how leaf N content altered under earthworm treatment. Consequently, it is unclear whether the changes in leaf N content induced changes in floral shoots or plant height in these two species? In addition, the authors also did show whether there was close linkage between community level increases in N concentration and NDVI increases under earthworm treatment. To my understanding, this kind of information is also crucial for understanding the major findings of this study. Without these evidences, it is hard to draw the major conclusion that 'earthworms unlock the arctic plant N limitation'.

Specific comments:

Line 27: 'Increased N availability resulted in increased fine root production'. I did not see any direct evidence to support this argument. Table S4 showed the weak associations between root growth and NH_4^+ , BUT Table S1 showed that worming effects did not alter soil N availability.

Line 32: "to a large extent". Why the authors say so? Low temperature is still an important factor.

Line 52: How general is the pattern of earthworm invasion?

Line 80 and thereafter: It is strange to use the term 'Ratio' after 15N?

Table 1: Please clearly show the statistical information so that the readers could easily understand the significance.

Figure 5: I am not sure how meaningful will this kind of comparison be, since the ecosystems may be different among various treatments. Also, the effect size may largely depend on the strength of treatment. For example, you can see that warming effects are totally different under 1 and 3 degree warming.

Line 262: Did the authors observe this kind of earthworms in your study site (Abisko Scientific Research Station)? How far from your study site and the city of Umea?

Line 269: "nearby", please show the specific distance. How about the possibility to observe 200 individual m⁻² in your study site (Abisko Scientific Research Station) and other arctic regions?

Line 282-289: Could the authors determine the rate of nutrient release from litter? Also, it might be better to show us the difference of litter (i.e., size and shape etc.) with and without earthworm treatment.

Line 321: The authors did not show plant community N in this section.

Reviewer #2 (Remarks to the Author):

Review of NCOMMS-19-17466 entitled: "Invasive earthworms unlock the arctic plant nitrogen limitation"

This manuscript is about effects of invasive earthworm species on N cycling and N uptake by shrub- and forb dominated vegetation tundra communities; specifically in two common tundra vegetation types, heath and meadow. The main finding is that invasive earthworms can increase N uptake (N concentration in plant biomass) in *Festuca ovina* by 50% and that in the presence of earthworms more N from litter is taken up by *F. ovina* (shown by 15N isotopic labelling of the litter). *F. ovina* was present in both vegetation types.

Other findings also point into the direction of increased plant productivity in the presence of invasive earthworms, e.g. increased %N in a shrub found in the heath vegetation and increased N uptake from litter in a forb from the meadow, both in the presence of invasive earthworms; and some more increases in plant height, floral shoots and root growth were found in the presence of earthworms.

In my view the topic of this manuscript is relevant, timely and of interest to other researchers in the field. The research settings (tundra soils and arctic plant growth) are novel and as far as I am aware this experimental work generates new knowledge.

The manuscript is generally well-written and with a good structure; in several cases rephrasing is needed or typos/spelling mistakes need to be repaired.

My main concerns are mostly about lack of clarity in the methodology and the way the results are used in the discussion and abstract. In the current manuscript these issues need to be clarified before I can recommend publication of this manuscript. I list my main comments/questions here below, and following these main comments I pointed out some more concrete comments concerning the results, discussion and methods sections.

Main comments (random order):

- Abstract:

- In general I think the abstract should be downtuned. For example:
- 'detritivores limitation' is probably not the only limitation for N availability. It's probably the lack of a bigger and more active soil food web, including detritivores such as earthworms.
- Also here would be good to mention the duration of the experiment
- L26: "...increased average plant N by as much as 50%". This holds only true for *F. ovina*. Here it reads like it holds true for the all vegetation. The same is true for the sentence to follow. All found effects are enlarged.

• L29-31: this sentence also reads like everything has been measured in this experiment, and that the comparison between worming, warming, herbivory and nutrient addition were done within one experiment.

- When introducing the topic, there is no information about earthworm effects on N cycling processes or plant growth outside arctic systems. For example, there's definitely information available about invasive earthworm effects on N dynamics as well as microbial activities for the Northern American and Canadian boreal forests.

- Is the information in L60-65 really necessary to introduce the topic? It could be discussed later in the discussion section ('A new paradigm for arctic soil N limitation').

- Some more experimental details are needed when introducing what has been done in the current experiment (L65-68): mention the two vegetation tundra communities; the duration of the experiment (103 days? One growing season? Not so clear to me)

- L68: "currently missing guilds in the decomposition process". "Guild" comes back a couple of times, and wouldn't "functional trait" be better? The use of plural for guild suggests not only one type of geoenvironmental engineer (such as earthworms) was used in the experiment. Even though I understand what is meant, I think "guild" needs different phrasing. It could be incorporated in the explanation of how earthworms can influence N dynamics in soils (see my first comment).

- Not explained in the Methods section:

- Greenness; how was that measured?
- NDVI (it is mentioned in L327, Methods section, but there is no explanation what NDVI data are and what the values mean)

- There are many dates mentioned in the Methods section; it becomes rather confusing to determine the exact duration of the experiment. Probably from June 13 to September 24 in 2018 (Figure 4). But there is also mention of June 2017 in L120 and L283. It could really help if some sort of visual timeline could be used for all the different procedures.

- It's not clear what part of the vegetation covers (heath and meadow) *F. ovina* covers. Is it substantial? The power of the results would become stronger if the increased N conc, such as in *F. ovina*, could be found for the total biomass as well.

- In this experiment it might be hard to claim that dead earthworms could not have contributed to the increased N availability and increased N conc in *F. ovina*, etc. because all the added worms died in the mesocosms during the first winter (L277). It could be worthwhile to estimate the amount of extra N that was added to these mesocosms.

- I find the first and second paragraph of the discussion (L126-154) too bold. Effects of earthworms on N uptake and plant growth have been shown to take place, but there is no proof for really dramatic changes in N cycling in arctic tundra systems. To make such claims I think more microbial N processes should have been measured (nitrification, denitrification, also microbial activity), and NO_3 and NH_4 should have been monitored instead of measured once at the end of the experiment. More comments on the discussion below.

More specific comments about the results:

- L95-103: effects in the presence or absence of earthworms are hard to distinguish here; this paragraph can be structured in a simpler way so it is immediately clear.

- Figure 2, graph on the right with $\Delta 15\text{N}$ on the y-axis, right panel about the meadow: is the average value for the 'with worms' treatment for *S. alpine* really correct? In that case it is hard to believe there is an effect found for 15N .

More specific comments about the discussion:

- L130: also in this place the word 'guild' reads strange. I suggest to rephrase that throughout the manuscript.
- L130-131: "substantial environmental change". Perhaps, but how exactly? This statement needs more explanation. Probably better to discuss this in the last subsection 'A new paradigm for arctic soil N limitation'.
- L132: As long as we don't know how substantial the earthworm effect on N conc in *F. ovina* is, it is too soon to say that earthworm activity greatly influences plant N uptake.
- L136: I am still at a loss with the "vegetation greenness".
- L139-141: sentence is not clear.
- L144-147: the measured effect of earthworm presence on fine root production does not become stronger because other studies about environmental factors had no effect on fine root production. I think the sentence needs to be rephrased. Even better to start comparing between earthworm effects and other environmental factors until the last paragraph of this first subsection of the Discussion (starting in L155).
- L159: Figure 5 looks rather convincing and it can be a good discussion, but I still don't understand the NDVI data.

More specific questions about the methodology:

- Was total aboveground biomass not recorded? And total root biomass?
- A mixture of endogeics and endo-epigeics was added: was the mixture always composed of the same number of earthworm species? Was the proportion always similar? This info is needed.
- What about earthworm survival after the experiment had finished?

Reviewer #3 (Remarks to the Author):

In this study, the authors investigate the effects of earthworms on N cycling in arctic soils, the premise being that invasive earthworms have the potential to spread across currently uninhabited areas of the Arctic and that they would fill a missing functional niche within these ecosystems. The authors used a mesocosm experiment with tundra heath and meadow plant communities in which half of the plots contained earthworms and the other half did not. They measured the effects of the earthworms on soil nutrients, microbial (bacterial + fungal) biomass, plant N uptake, plant production of fine roots and shoots, and NDVI. They found that while earthworms did not seem to affect soil nutrient pools, they did affect N content of plants and plant growth in both heath and meadow systems. The authors also compare the magnitude of the earthworm effects they document to other global change drivers (e.g., warming, herbivory, fertilization) from previously published papers and make the argument that earthworms have the potential to have large effects on arctic N cycling.

The authors have identified an interesting question – given the risk of earthworm invasion to certain areas of the Arctic, what will be the consequences for N cycling? The questions and the study are topical and interesting, and I appreciate the amount of work that went into conducting a study like this in the Arctic. It was a well-designed experiment that seemed to be carefully conducted, and the results, specifically in regards to plant N content and NDVI, are compelling. However, some revisions are necessary with regards to the presentation of the results and in order to more effectively link the many different responses that were measured in the experiment.

For example, the effects of the earthworms appear to be different in the meadow and the heath vegetative communities but it is unclear why (e.g., worms increased plant N more in the heath, worms reduced bacterial biomass in the meadow but not the heath, etc). The importance of detritivores has been linked in many different studies to the availability of moisture –this may help explain some of the differences between the heath and meadow communities. From Fig. S1, it looks like soil moisture was

only collected in a few of the plots but there should have been a measure of soil moisture from the PFLA work (as indicated in line 301) that could be included in the models.

Likewise, the authors measured fungal and bacterial biomass but only report on worm effects on bacterial biomass specifically in the results section. This was surprising to me, because although the worms did not affect the fungi:bacteria ratio in either the heath or the meadow, the results from Table 1 suggest that the addition of worms decreased bacterial biomass in the meadow but increased fungal biomass in the heath. This points to potentially different effects of worm activity in the two tundra types but the full model results for fungi are not reported.

Along the same lines, most of the results of the linear mixed effects models upon which the paper is based are only partially reported on in the text. It would help readers to be able to interpret and assess the results if there was a series of tables that showed the full output from all models (in the supplement if space does not allow in the main text).

With regards to the earthworms themselves, the authors argue that there is a risk of these organisms spreading throughout those parts of the Arctic with discontinuous permafrost. Yet it is also mentioned in the methods that the earthworms could not survive the winter over the course of the experiment, so there is a disconnect regarding whether they can or cannot exist in the harsh arctic conditions. In areas where earthworms have been transported, are they able to survive the winter and actually become established, or is it that they are constantly reintroduced by human activity? Is the study site further north with harsher winters than areas where earthworms have become established? This is unclear from the text but is relevant to the "invasion debt" argument outlined in the introduction.

Specifically, the authors make the argument that earthworms may have an "invasion debt" whereby they could survive in certain areas but have been limited by slow dispersal (lines 60-65). There is no mention of whether there is historical evidence to suggest that earthworms – or another large detritivore counterpart that would have filled this functional role -- existed in the Arctic in the past, but it would be a nice addition to briefly touch on for some context.

There is a rich body of literature on the geoengineering effects of invasive earthworms (e.g., referenced in the introduction; line 51). In the discussion, it would be worthwhile to briefly compare how or why the effects of the earthworms in this experiment differ from those in other systems where they have invaded.

Lines 155-171: I appreciate that the authors compared the earthworm effects on plant N and NDVI to those of other major global change drivers in the Arctic (fertilization, warming, herbivory, shrubification). This approach drives home the message that these missing detritivores could have important effects on arctic ecosystems. However, this figure would be even more convincing if error bars could be added. More explanation is also needed for the figure caption and in the supplement regarding the data that were used and how the figure was generated. For example, it is unclear whether the herbivory panels show the effects of herbivore presence or exclusion.

Other comments:

- Is *F. ovina* a dominant plant in this area of the Arctic? Was the emphasis on it simply because it existed in both the heath and the meadow or does it have other significance?

- Mentioning a few key methods-related details earlier on would make interpretation of the results and discussion easier. For example, results are presented from days 55 and 103 after the start of the experiment (lines 123 and 103) but the total length of the experiment hadn't been indicated yet. Noting this earlier would be helpful since the length of the experiment is so relevant in places like the Arctic where growing seasons are short and things generally happen slowly. The change in N content and growth of the plants that the authors found over such a short period is actually quite striking but that wasn't apparent until after reading the methods section. It could be highlighted earlier.

- Lines 156, 160, etc: Please define community-N. Is this average N content of the whole plant community? It wasn't clear from the results section or the supplement.

- It is unclear where the PFLA analyses for the soils came from. Was this microbial biomass from the litter layer or the upper organic soil layer? I could also not find how litter cover at the end of the experiment was assessed. I assumed litter cover was in reference to the litter addition but I am not sure how one would do that since the labeled/unlabeled litter that was added was ground litter, which would be nearly impossible to quantify after several months.

- Lines 321-326 describe the methods for assessing plant community composition and the abundance of the various plant functional groups within the mesocosms but I didn't see that this information was reported anywhere other than in vague terms (e.g., that *F. ovina* was present in heath and meadow plots).

Reviewer #1 (Remarks to the Author):

COMMENT 1# First, the authors should provide strong evidence that the invasion of earthworms could widely exist in arctic ecosystems, which is the basis for the current study. How frequent will the earthworms be observed in current and future climate scenario? How about the invasion rate of earthworm across the arctic region? How general is the pattern of earthworm invasion? Do the authors have some actual field observations, or model predictions? This issue is crucial for current study, and the authors must show these background data so as to help the readers understand the principle of current study. BTW, to my experience on the Tibetan Plateau, we have very few chances to observe earthworms in this largest alpine permafrost region of the Northern Hemisphere.

Reply: To provide strong evidence that earthworms can survive in many Arctic ecosystems we have:

- 1) Added a map that illustrates published observations about non-native geoeengineering earthworm populations within arctic environments (Fig 1). As illustrated in this figure, geoeengineering earthworms have established (via human transport vectors) in the Scandinavian arctic, Greenland, Iceland, the North American (USA and Canada) and Russian arctic. This clearly supports the view that bioclimatic conditions currently existing within parts of the Arctic allow the establishment of these earthworm species.
- 2) Cited a paper in press in *Science* (Philips et al 2019), which uses both soil properties and climatic variables to demonstrate that suitable earthworms habitats extends well into arctic regions (line 61-62).
- 3) Explained that warmer soil in combination with intensified human land-use will facilitate earthworm expansions in the near future (lines 58-59). For example, increased soil temperatures and formation of year-around unfrozen soil layers (talik) as predicted for arctic environments (Nicolosky et al 2017) will open up new niches where soil frost has been limiting establishments. At the end of the introduction, we further discuss findings from a recent field study (Wackett et al. 2018) demonstrating that our target earthworm species has been found to thrive in and re-shape the morphology of arctic soils at sites adjacent to our experiment (lines 71-73).

Regarding the invasion rate we have explicitly stated in the introduction that earthworms invade landscapes over decadal to centennial timescale, i.e. the same time-scale as climate models foresee climate change (lines 52-53). We further expand the contents of the new map and explain that human activities have also induced abrupt colonization events across the Arctic (lines 54-55) shown in Fig 1.

COMMENT 2# Second, the authors should provide more evidence that earthworms could alter N cycling processes in arctic ecosystems, which could help to understand why plant N concentration increased under 'earthworm' treatment. Although the authors showed that worming effects led to the elevated leaf N concentrations in several species, it is still unclear what kind of ecological processes drove this N increase. Particularly, Table S1 revealed that worming effects did not alter soil N availability. To my deduction, the possible reason for the increased leaf N concentration is that earthworms stimulated nutrient release from litter or soil organic matter mineralization/nitrification? However, there is no evidence whether and how worming effects altered these two key processes. I

strongly suggest to measure nutrient release rate from litter, and also determine soil nitrogen transformation processes (using N isotope technique) with and without worms treatment. These data are key for understanding the patterns observed in this study.

Reply: We have now additionally measured N release from earthworm casts and identified that earthworms increase N mineralization and that nitrification is a minute process in our soils (L 114-118, 145-147, 190-195, FigS3). In the revised text we have clarified how our ¹⁵N isotopic study support that the plants get a higher access to N from aboveground litter in the presence of earthworms (lines 195-204). We have also clearly stated that elevated leaf N concentrations in several species without any measurable increase of nutrient availability in the soil is corroborating with other experimental and field studies in the Arctic (line 212-216). That is, arctic plants are extremely competitive for N and increased concentrations of N forms in the soil solution rarely occur in response to increased N availability. More detailed mechanistic measurements of soil processes is beyond the scope of this study as our study does not target N cycling but rather overall earthworm colonization impacts on plant N uptake.

COMMENT 3# Third, the authors should provide strong evidence to illustrate the linkage between changes in leaf nutrient and plant growth dynamics under earthworm treatment. Currently, the authors observed the increases in leaf N content and floral shoots of *Festuca ovina*. Is there any association between the increases in floral shoots and leaf N content of *Festuca ovina*? Also, the authors observed the increases in plant height of *Deschampsia fleuosa*, but did not show us whether and how leaf N content altered under earthworm treatment. Consequently, it is unclear whether the changes in leaf N content induced changes in floral shoots or plant height in these two species? In addition, the authors also did show whether there was close linkage between community level increases in N concentration and NDVI increases under earthworm treatment. To my understanding, this kind of information is also crucial for understanding the major findings of this study. Without these evidences, it is hard to draw the major conclusion that 'earthworms unlock the arctic plant N limitation'.

Reply: In the previous version of the manuscript we focused on the effects of the experimental treatments since these is the ones that reveal a causal relationship between earthworms and plant N concentrations, and growth above and belowground. Yet, we do agree with the reviewer that relationships between response variables can strengthen the discussion about how plausible mechanisms behind these effects of earthworms are (in our case demonstrating that increased uptake of N drive observed growth responses). As suggested by the reviewer, we now provide statistics for that N-content is positively related to the number of floral shoots ($r^2 = 0.518$, $P = 0.001$) in *Festuca ovina* (lines 129-132). We show that there is a similar positive relationship between the N-content in *Festuca ovina* and the NDVI of the plant community ($r^2 = 0.487$, $P = 0.002$). Note that there is also a positive correlation N-content in *Festuca ovina* and community-N ($r^2 = 0.386$, $P = 0.017$), but we prefer not to include this information as the link between plant N and plant growth is already indicated (by correlations above) and as this measure is only used in the synthesis (Fig 6) and not a fully independent measure (N content of *F ovina* is used in the calculation of community N). N content was not measured for *Deschampsia*

flexuosa, since it was not found in all mesocosms, and thus similar analyses cannot be done for this species.

Specific comments:

COMMENT 4# Line 27: Increased N availability resulted in increased fine root production. I did not see any direct evidence to support this argument. Table S4 showed the weak associations between root growth and NH₄⁺, but Table S1 showed that worming effects did not alter soil N availability.

Reply: We agree with the reviewer that we have no direct evidence of a causal relationship between root growth and N availability and have this rephrased this in the manuscript and removed Table S4.

COMMENT 5# Line 32: “to a large extent”. Why the authors say so? Low temperature is still an important factor.

Reply: We have replaced this sentence with “Here, we show that arctic plant-soil N cycling is also substantially constrained by the lack of larger detritivores (earthworms) able to mineralize and physically translocate litter and soil organic matter” Line 24-26.

COMMENT 6# Line 52: How general is the pattern of earthworm invasion?

Reply: We have added information about the circumpolar presence of earthworm populations in sub-arctic and low-arctic environments by adding a map with some known finds in North America (USA and Canada), Greenland, Iceland, Fennoscandia and Russia (Fig 1). We are convinced that this map clearly illustrate that geoengineering earthworms are establishing in arctic environments. Only records were used for earthworm populations that are considered to be non-native (encompassing the species used in this study and the endogenic species *Apporectodea caliginosa*) and where specific locations have been specified in the publications.

COMMENT 7# Line 80 and thereafter: It is strange to use the term ‘Ratio’ after 15N?

Reply: Thanks for highlighting this. In the revised version, we now use consistently $\delta^{15}\text{N}$.

COMMENT 8# Table 1: Please clearly show the statistical information so that the readers could easily understand the significance.

Reply: The required information is added to Table 1. Full statistical summary is added in Table S5.

COMMENT 9# Figure 5: I am not sure how meaningful will this kind of comparison be, since the ecosystems may be different among various treatments. Also, the effect size may largely depend on the strength of treatment. For example, you can see that warming effects are totally different under 1 and 3 degree warming.

Reply: Please note that this comparison is based solely on local experiments which explicitly minimizes the bias caused by regionally differences in conditions. In the revised version of the ms. we have more clearly presented the local aspect of this comparison (lines 171-173). We have also added standard error bars (in line with comment 45) to indicate uncertainties of the estimates.

Opposite to Reviewer 1, Reviewers 2 (comment #35) and 3 (comment #45) are specifically in favor of this comparison and thus, we consider the comparison informative.

COMMENT 9# Line 262: Did the authors observe this kind of earthworms in your study site (Abisko Scientific Research Station)? How far from your study site and the city of Umea?

Reply: Yes, we did and we have now specified that there are four known arctic invasion sites within a radius of 12 km from our mesocosm experiment (line 295-298)

COMMENT 10# Line 269: “nearby”, please show the specific distance. How about the possibility to observe 200 individual m⁻² in your study site (Abisko Scientific Research Station) and other arctic regions?

Reply: We have specified the distance (<12 km, 298) and that abundances at these sites reach 200 individuals m⁻² and thus directly comparable to densities used in our experiment (line 303-305).

COMMENT 11 # Line 282-289: Could the authors determine the rate of nutrient release from litter? Also, it might be better to show us the difference of litter (i.e., size and shape etc.) with and without earthworm treatment.

Reply: We have added measurements of leachable N forms from litter (Fig S3). Nutrient release from litter is not measured *per se* in the mesocosm, but by adding spiked litter we have been able to follow the fate of N from litter in the mesocosms. The fine texture of earthworm cast is evident in Fig S2.

COMMENT 12# Line 321: The authors did not show plant community N in this section.

Reply: Correct, this was a mistake. We have removed this from the section title.

Reviewer #2 (Remarks to the Author):

In my view the topic of this manuscript is relevant, timely and of interest to other researchers in the field. The research settings (tundra soils and arctic plant growth) are novel and as far as I am aware this experimental work generates new knowledge. The manuscript is generally well-written and with a good structure; in several cases rephrasing is needed or typos/spelling mistakes need to be repaired.

Reply: During the revision process, we have paid attention to correct typos and spelling mistakes.

My main concerns are mostly about lack of clarity in the methodology and the way the results are used in the discussion and abstract. In the current manuscript these issues need to be clarified before I

can recommend publication of this manuscript. I list my main comments/questions here below, and following these main comments I pointed out some more concrete comments concerning the results, discussion and methods sections.

Reply: We have revised the methodology, results, discussion and abstract following the reviewer's more detailed comments below. For instance, we now provide full reports for all statistical analysis in tabulated form in the supplements and refer to these accordingly in the plain/main text.

COMMENT #13 In general I think the abstract should be downtuned. For example:

'Detritivores limitation' is probably not the only limitation for N availability. It's probably the lack of a bigger and more active soil food web, including detritivores such as earthworms. Also here would be good to mention the duration of the experiment

Reply: We removed 'detritivore limitation' and added information about the length (2 years) of the study in line with the reviewer comment (line 27).

COMMENT #14 L26: "...increased average plant N by as much as 50%". This holds only true for F. ovina. Here it reads like it holds true for the all vegetation. The same is true for the sentence to follow. All found effects are enlarged.

Reply: We now specify that both shrubs and grass increased their N content, but avoid specifying the average 50% increase that was specific for *F. ovina*

COMMENT #15 L29-31: this sentence also reads like everything has been measured in this experiment, and that the comparison between worming, warming, herbivory and nutrient addition were done within one experiment.

Reply: In the revised abstract, we specify that the studies with warming, herbivory and nutrient addition, which we used to compare our 'worming' effects with, were reported in previous studies (line 30-33).

COMMENT #16 When introducing the topic, there is no information about earthworm effects on N cycling processes or plant growth outside arctic systems. For example, there's definitely information available about invasive earthworm effects on N dynamics as well as microbial activities for the Norther N American and Canadian boreal forests.

Reply: We have added references regarding known effects on N cycling processes in other biomes and their impacts on N transformation, microbial processes and plant growth (line 63-67).

COMMENT #17 Is the information in L60-65 really necessary to introduce the topic? It could be discussed later in the discussion section ('A new paradigm for arctic soil N limitation').

Reply: Reviewer #1 requested more information about the dispersal rates and likelihoods of earthworms to colonize the Arctic, as this was considered essential to justify our study. However, we have rephrased this section to make it more clear. We think this information is of high conceptual importance

highlighting that the current low abundances of earthworms at high latitudes is likely an artefact of slow migration rates rather than due to current harsh climatic conditions.

COMMENT #18 Some more experimental details are needed when introducing what has been done in the current experiment (L65-68): mention the two vegetation tundra communities; the duration of the experiment (103 days? One growing season? Not so clear to me)

Reply: We added a table illustrating the time-line of different activities (manipulations, sampling and measurements) of the experiment (Table S1). Information about the vegetation types (heath and meadow) and the length (2 years) of the study is also added (line 68-71).

COMMENT #19 L68: “currently missing guilds in the decomposition process”. “Guild” comes back a couple of times, and wouldn’t “functional trait” be better? The use of plural for guild suggests not only one type of geoenvironmental engineer (such as earthworms) was used in the experiment. Even though I understand what is meant, I think “guild” needs different phrasing. It could be incorporated in the explanation of how earthworms can influence N dynamics in soils (see my first comment).

Reply: We replaced “guild” with “functions” throughout the ms.

COMMENT #20-22 Not explained in the Methods section: Greenness; how was that measured? NDVI (it is mentioned in L327, Methods section, but there is no explanation what NDVI data are and what the values mean)

Reply: NDVI (Normalized Difference Vegetation Index) is a common measure of vegetation “Greenness”, and we have explained these two terms and this measurement more clearly in the revised version of the ms (lines 121-122, line 144, line 171).

COMMENT #23 There are many dates mentioned in the Methods section; it becomes rather confusing to determine the exact duration of the experiment. Probably from June 13 to September 24 in 2018 (Figure 4). But there is also mention of June 2017 in L120 and L283. It could really help if some sort of visual timeline could be used for all the different procedures.

Reply: We added a table illustrating the time-line of different activities (manipulations, sampling and measurements) of the experiment (Table S1).

COMMENT #24 It’s not clear what part of the vegetation covers (heath and meadow) *F. ovina* covers. Is it substantial? The power of the results would become stronger if the increased N conc, such as in *F. ovina*, could be found for the total biomass as well.

Reply: The proportional cover of vegetation are shown in Table S2 and the total biomass weighted N-content (community-N) is shown in Fig 6a. We measured the N% in 5 species. In the two dwarf shrubs and the graminoid (*F. ovina*), worms increased N concentrations substantially. Together these functional groups make up the majority of plant biomass in both vegetation types. N concentrations in the two forb species did however not increase by worms, and forbs are an important component of the meadows.

COMMENT #25 In this experiment it might be hard to claim that dead earthworms could not have contributed to the increased N availability and increased N conc in *F. ovina*, etc. because all the added worms died in the mesocosms during the first winter (L277). It could be worthwhile to estimate the amount of extra N that was added to these mesocosms.

Reply: We measured N availability and N content in plants in the first summer before the presumed mortality in winter and hence we think it is reasonable to assume that the plant N uptake is not due to dead worms. Nevertheless, we acknowledge this source but list two arguments why this source is negligible (line 204-209).

COMMENT #26 I find the first and second paragraph of the discussion (L126-154) too bold. Effects of earthworms on N uptake and plant growth have been shown to take place, but there is no proof for really dramatic changes in N cycling in arctic tundra systems. To make such claims I think more microbial N processes should have been measured (nitrification, denitrification, also microbial activity), and NO₃ and NH₄ should have been monitored instead of measured once at the end of the experiment. More comments on the discussion below.

Reply: We have now rephrased these paragraphs to make them less bold, but also added new measurements on N transformation processes (Fig S3).

COMMENT #27 L95-103: effects in the presence or absence of earthworms are hard to distinguish here; this paragraph can be structured in a simpler way so it is immediately clear.

Reply: The section has been restructured and rephrased to emphasize effects from earthworms.

COMMENT #28 Figure 2, graph on the right with delta 15N on the y-axis, right panel about the meadow: is the average value for the 'with worms' treatment for *S. alpine* really correct? In that case it is hard to believe there is an effect found for 15N.

Reply: We thank the reviewer for spotting this graphical error. It has been fixed in the revision version of the ms.

More specific comments about the discussion:

COMMENT #29 L130: also in this place the word 'guild' reads strange. I suggest to rephrase that throughout the manuscript.

Reply: Done (see response to comment #19)

COMMENT #30 L130-131: "substantial environmental change". Perhaps, but how exactly? This statement needs more explanation. Probably better to discuss this in the last subsection 'A new paradigm for arctic soil N limitation'.

Reply: Discussion moved to the last subsection in line with this comment and rephrased accordingly.

COMMENT #31 L132: As long as we don't know how substantial the earthworm effect on N conc in *F. ovina* is, it is too soon to say that earthworm activity greatly influences plant N uptake.

Reply: This comment seems to be related to the slight misconception shown in comment #24. That is, the effect on N concentration for *F.ovina* and the other four plants with either increased N concentrations or changed N sources (seen by their $\delta^{15}\text{N}$ composition) in response to earthworms, is indeed substantial even at the community level in comparison to other drivers of environmental change (Fig 6a). However, we have avoided the use of 'greatly' when referring to the impact on plant N uptake (Line 140-142) prior to the comparisons when it becomes evident that the impact is indeed great (line 173-176).

COMMENT #32 L136: I am still at a loss with the “vegetation greenness”.

Reply: To clarify what we mean with vegetation greenness, we have added a description in the material and method section explaining that the NDVI is used commonly as a measure of vegetation greenness (Line 368-371) as well as in the new text (in brackets) in the discussion that the NDVI is considered a measure of greenness (lines 121-122, line 144, line 171).

COMMENT #33 L139-141: sentence is not clear.

Reply: This sentence has now been rephrased.

COMMENT #34 L144-147: the measured effect of earthworm presence on fine root production does not become stronger because other studies about environmental factors had no effect on fine root production. I think the sentence needs to be rephrased. Even better to start comparing between earthworm effects and other environmental factors until the last paragraph of this first subsection of the Discussion (starting in L155).

Reply: This sentence has been rephrased and the section is now moved as suggested by the reviewer.

COMMENT #35 L159: Figure 5 looks rather convincing and it can be a good discussion, but I still don't understand the NDVI data.

Reply: We have now added an explanation (in brackets) for NDVI in this figure caption. In addition to this reply, please see our responses to comment #20 and #32.

COMMENT #36 Was total aboveground biomass not recorded? And total root biomass?

Reply: These measurements demands harvesting the mesocosm, which is not done since we still like to study more long-term effects. However, our pinpoint measure of plant coverage (Table S2) and NDVI are proxies for biomass, and total root growth can be inferred from the measurements shown in Fig 4.

COMMENT #37 A mixture of endogeics and endo-epigeics was added: was the mixture always composed of the same number of earthworm species? Was the proportion always similar? This info is needed.

Reply: Yes. This information has been added to the methods.

COMMENT #38 What about earthworm survival after the experiment had finished?

Reply: Again, the experiment is still running, thus we cannot provide data that requires destructive harvesting.

Reviewer #3 (Remarks to the Author):

COMMENT #39 The authors have identified an interesting question – given the risk of earthworm invasion to certain areas of the Arctic, what will be the consequences for N cycling? The questions and the study are topical and interesting, and I appreciate the amount of work that went into conducting a study like this in the Arctic. It was a well-designed experiment that seemed to be carefully conducted, and the results, specifically in regards to plant N content and NDVI, are compelling. However, some revisions are necessary with regards to the presentation of the results and in order to more effectively link the many different responses that were measured in the experiment.

For example, the effects of the earthworms appear to be different in the meadow and the heath vegetative communities but it is unclear why (e.g., worms increased plant N more in the heath, worms reduced bacterial biomass in the meadow but not the heath, etc). The importance of detritivores has been linked in many different studies to the availability of moisture –this may help explain some of the differences between the heath and meadow communities. From Fig. S1, it looks like soil moisture was only collected in a few of the plots but there should have been a measure of soil moisture from the PFLA work (as indicated in line 301) that could be included in the models.

Reply: We now present that there were no earthworm effect on the soil moisture (line 100-103). We acknowledge that differences in microbial community responses may be due to a more fierce plant competition for N during the summer in the meadow tundra as this plant community show an earlier response in fine root growth (Line 218-223),

COMMENT #40 Likewise, the authors measured fungal and bacterial biomass but only report on worm effects on bacterial biomass specifically in the results section. This was surprising to me, because although the worms did not affect the fungi:bacteria ratio in either the heath or the meadow, the results from Table 1 suggest that the addition of worms decreased bacterial biomass in the meadow but increased fungal biomass in the heath. This points to potentially different effects of worm activity in the two tundra types but the full model results for fungi are not reported.

Statistically significant effects of worms were found only for Bacteria and *Actinomycetes*. This is now clearly shown in Table 1.

COMMENT #41 Along the same lines, most of the results of the linear mixed effects models upon which the paper is based are only partially reported on in the text. It would help readers to be able to interpret and assess the results if there was a series of tables that showed the full output from all models (in the supplement if space does not allow in the main text).

Reply: We have added a tables summarizing the full output of the linear mixed model (Table S5)

COMMENT #42 With regards to the earthworms themselves, the authors argue that there is a risk of these organisms spreading throughout those parts of the Arctic with discontinuous permafrost. Yet it is also mentioned in the methods that the earthworms could not survive the winter over the course of the experiment, so there is a disconnect regarding whether they can or cannot exist in the harsh arctic conditions. In areas where earthworms have been transported, are they able to survive the winter and actually become established, or is it that they are constantly reintroduced by human activity? Is the study site further north with harsher winters than areas where earthworms have become established? This is unclear from the text but is relevant to the “invasion debt” argument outlined in the introduction.

Reply: Earthworm can indeed survive and even thrive in the local environment of our study site (see Wackett et al 2018), and elsewhere in the Arctic (see new Fig. 1). However, the mesocosm are constructed in a way that do not allow behavioral responses to soil frost, i.e. earthworm in the mesocosm cannot migrate away from the soil frost in similarity to natural populations because of a plastic barrier at 30 cm depth. We have stated this more clearly in the revised version of the ms (line 310-315).

COMMENT #43 Specifically, the authors make the argument that earthworms may have an “invasion debt” whereby they could survive in certain areas but have been limited by slow dispersal (lines 60-65). There is no mention of whether there is historical evidence to suggest that earthworms – or another large detritivore counterpart that would have filled this functional role -- existed in the Arctic in the past, but it would be a nice addition to briefly touch on for some context.

Reply: No such evidence besides that cited already in the paper exist for the arctic. With the new Fig. 1 illustrating presence of geoengineering earthworms populations at various arctic environments we show that this group of detritivores can indeed survive at high latitudes once they have been able to spread there. That these earthworms can establish via such human assisted long-distance dispersal is a strong evidence for that the vast majority of land in the arctic is open for earthworm and thus, that this landscape has a considerable ‘invasion debt’ due to slow dispersal rates north. Note that this part of the introduction has been modified in response to suggestions by reviewer 1 and 2 (comment #1 and 17)

COMMENT #44 There is a rich body of literature on the geoengineering effects of invasive earthworms (e.g., referenced in the introduction; line 51). In the discussion, it would be worthwhile to briefly compare how or why the effects of the earthworms in this experiment differ from those in other systems where they have invaded.

Reply: We have added comparisons with other systems when appropriate in the discussion (Line 155-158, 163-165, 194-195).

COMMENT #45 Lines 155-171: I appreciate that the authors compared the earthworm effects on plant N and NDVI to those of other major global change drivers in the Arctic (fertilization, warming, herbivory, shrubification). This approach drives home the message that these missing detritivores could have important effects on arctic ecosystems. However, this figure would be even more

convincing if error bars could be added. More explanation is also needed for the figure caption and in the supplement regarding the data that were used and how the figure was generated. For example, it is unclear whether the herbivory panels show the effects of herbivore presence or exclusion.

Reply: Error bars have been added and the figure caption expanded in line with the reviewer comment. An equation has been added in the supporting information to explain how the community N was calculated.

Other comments:

COMMENT #46 Is *F. ovina* a dominant plant in this area of the Arctic? Was the emphasis on it simply because it existed in both the heath and the meadow or does it have other significance?

Reply: *Festuca ovina* was the only species common enough in both the heath and the meadow to sample in all mesocosms and this is now clearly stated in the manuscript.

COMMENT #47 Mentioning a few key methods-related details earlier on would make interpretation of the results and discussion easier. For example, results are presented from days 55 and 103 after the start of the experiment (lines 123 and 103) but the total length of the experiment hadn't been indicated yet. Noting this earlier would be helpful since the length of the experiment is so relevant in places like the Arctic where growing seasons are short and things generally happen slowly. The change in N content and growth of the plants that the authors found over such a short period is actually quite striking but that wasn't apparent until after reading the methods section. It could be highlighted earlier.

Reply: Thank you for pointing this out. We have added the total duration in the abstract (line 27) and introduction (line 69), and also clearly point out the fast responses in the discussion (for example line 142). A timeline of the experiment is also shown in Table S1.

COMMENT #48 Lines 156, 160, etc: Please define community-N. Is this average N content of the whole plant community? It wasn't clear from the results section or the supplement.

Reply: Explanation in the text has been added in brackets (+ equation in the supporting information).

COMMENT #49 It is unclear where the PFLA analyses for the soils came from. Was this microbial biomass from the litter layer or the upper organic soil layer? I could also not find how litter cover at the end of the experiment was assessed. I assumed litter cover was in reference to the litter addition but I am not sure how one would do that since the labeled/unlabeled litter that was added was ground litter, which would be nearly impossible to quantify after several months.

Reply: For the PFLA, the samples included both the litter layer and the organic soil layer, which we explain in the revised version of the ms. The litter cover was measured using the point intercept method (unrelated to the added, labeled litter). To make it more clear, we have added litter in the method subheading that now reads: "Aboveground plant composition, production and litter cover"

Manuscript **NCOMMS-19-17466A**
From Jonatan Klaminder
Department of Ecology and Environmental Science
Umeå University
Jonatan.klaminder@umu.se

COMMENT #50 Lines 321-326 describe the methods for assessing plant community composition and the abundance of the various plant functional groups within the mesocosms but I didn't see that this information was reported anywhere other than in vague terms (e.g., that *F. ovina* was present in heath and meadow plots).

Reply: This information is reported in Table S2.

Reviewers' comments:

Reviewer #1 (Remarks to the Author):

The authors have done a good job to address my previous comments. They provided a map to show the distribution of earthworms across the arctic region. They also provided some indirect evidence to show earthworm could enhance microbial mineralization process, although I expect to see some direct measurements about changes in litter/soil mineralization and nitrification processes with and without worms treatment. They further showed the close linkage between changes in leaf nutrient and plant growth dynamics under earthworm treatment. Overall, I am satisfied with the revision and have no further major concerns.

Reviewer #3 (Remarks to the Author):

The topic of this manuscript is relevant and should be interesting to both arctic and non-arctic researchers alike. Overall, the authors present ample evidence from a well-designed study showing that earthworms have a variety of effects related to N cycling, and I think that this would be a novel contribution to the literature. By providing background information on the risk of earthworm invasion in the North, the revised version of this manuscript does a good job of justifying why understanding earthworm impacts on tundra ecosystems is timely. My main concerns are that the results section is missing some pertinent information and there are a few places where results seem to be overstated. I also think that the introduction would benefit from some restructuring (please see specific line comments below). Additionally, there are several typos and sentences that need clarifying within the text.

Specific comments:

Line 27: Typo: no apostrophe on earthworms

Line 27: "Two-year-long" should be over two growing seasons.

Line 29: Does earthworm activity increase these plant traits in both habitats? It would be helpful to indicate that. I see what the authors are doing here but it's not entirely clear in the abstract whether the changes observed exceeded the differences between shrub and forb communities only in the shrub habitat, only with the forbs or both. Because the main text refers to the habitat types as heath and meadow, it's also not clear whether this reference to shrub and forb communities are the experimental heath and meadow plots or whether they're in reference to other general tundra sites.

Line 34: Typo: no 's' in plays

Line 36: Typo: add an 's' to play

Line 42: I think it would be helpful to add a phrase here differentiating abiotic drivers of changes in N cycling (e.g., warming or external inorganic N inputs) vs. potential biotic drivers (e.g., herbivory) as a way to lead into the role of soil macrofauna.

Lines 49-52: This sentence should be rephrased; it might help this section to add a sentence in line 49-50 first stating that many areas experienced eradication of soil macrofauna during glacial periods and then discuss the natural vs. human mediated dispersal issue.

Lines 63-65: This is a nice summary of how earthworms change N cycling in other ecosystems. It seems like it should go earlier (e.g., at the beginning of the second paragraph or in line 48) in order to introduce the types of impacts they have so readers can see immediately why their presence in the

Arctic could modify N dynamics.

Lines 72-73: Consider introducing the study species before the experiment as a way of justifying why those species were used.

Lines 68-71: The authors might consider rephrasing or splitting this sentence into two.

Line 69: According to Table S1, the experiment ran from June 2017-October 2018, which would make it over two growing seasons and not over two full years. This is relevant, as winter dynamics could also be important for nutrient dynamics. Moreover, the winter season wouldn't count anyway because presumably all the earthworms died and had to be restocked during the next growing season.

Line 80: The next section gives results for effects of earthworms on plant production after one and two growing seasons but it's not clear whether the analyses of plant N concentration, N sources, litter cover, PRS probes, PLFA only took place after the first season or both (line 80 refers to plant N and N sources after one growing season but it isn't mentioned for the second season or for the other analyses in this section).

Line 90: This sentence seems to be somewhat misleading. Of the four species-specific results described in this paragraph, there were significant effects of earthworms on N concentrations only in one of them (*Vaccinium vitis-idaea*), which was in the heath. There doesn't seem to be evidence that the increased plant N came from the litter layer in the heath, because that result was in *Saxifraga alpina*, a forb in the meadow. Moreover, the comment in lines 91-92 suggesting that there were relatively larger effects of an increase in N concentration in the heath should be rephrased if in fact the one heath plant was the only one to show significant changes in plant N (as worded, it sounds as though there were changes in the meadow but that the changes were larger in the heath). The heath-specific ^{15}N results are also missing from this paragraph.

Lines 90-99: Were interactions with litter labeling tested for all species but only found to be significant for *S. alpina*? This result should be added for the heath species.

Line 101: Typo - open parenthesis missing

Line 103: Since it hasn't been mentioned before, it would be helpful to have an opening sentence here indicating that analyses of soil nutrients and microbial biomass were done using PRS and PLFA with very brief explanation of the acronyms.

Line 106-107: Was the vertical mixing in both habitats?

Line 107: Likewise, it would be helpful to indicate here whether microbial biomass was measured from soil or litter samples.

Line 109. Is the fungal biomass result a typo (meant to be $P > 0.3579$)?

Line 139: Overall, the discussion is well-written and does a good job of placing the results in the context of the other drivers of change occurring in the Arctic.

Lines 211-212: Given that the authors did not record increases in mineral N in the soil solution, I'd suggest toning down this sentence to saying that the mobilization of N from litter and availability of N in earthworm casts were associated with higher plant N content rather than that they resulted in higher plant N.

Line 218: typo in this line

Line 245: typo - no 's' in exists

Line 249: Given that the experiment only lasted 2 years, the suggestion that the results show that earthworm effects could be long-lasting and potentially irreversible seems like an overreach.

Point by point reply to the reviewer #3 suggested edits

Reviewer #3 (Remarks to the Author):

Comment 1. Line 27: Typo: no apostrophe on earthworms

Reply: Typo has been corrected

Comment 2. Line 27: “Two-year-long” should be over two growing seasons.

Reply: Now reads “over two growing seasons”

Comment 3. Line 29: Does earthworm activity increase these plant traits in both habitats? It would be helpful to indicate that. I see what the authors are doing here but it’s not entirely clear in the abstract whether the changes observed exceeded the differences between shrub and forb communities only in the shrub habitat, only with the forbs or both. Because the main text refers to the habitat types as heath and meadow, it’s also not clear whether this reference to shrub and forb communities are the experimental heath and meadow plots or whether they’re in reference to other general tundra sites.

Reply: Yes, plant traits are affected in both habitats, but the effects are on different traits (height and number of floral shoots). We have rephrased to make this clear.

Comment 4. Line 34: Typo: no ‘s’ in plays

Reply: Typo have been corrected

Comment 5. Line 36: Typo: add an ‘s’ to play

Reply: Typo have been corrected

Comment 6. Line 42: I think it would be helpful to add a phrase here differentiating abiotic drivers of changes in N cycling (e.g., warming or external inorganic N inputs) vs. potential biotic drivers (e.g., herbivory) as a way to lead into the role of soil macrofauna.

Reply: We have rephrased in line with the reviewer comment

Comment 7. Lines 49-52: This sentence should be rephrased; it might help this section to add a sentence in line 49-50 first stating that many areas experienced eradication of soil macrofauna during glacial periods and then discuss the natural vs. human mediated dispersal issue.

Reply: We have rephrased in line with the reviewer comment

Comment 8. Lines 63-65: This is a nice summary of how earthworms change N cycling in other ecosystems. It seems like it should go earlier (e.g., at the beginning of the second paragraph or in line 48) in order to introduce the types of impacts they have so readers can see immediately why their presence in the Arctic could modify N dynamics.

Reply: We have moved this section in line with the reviewer comment

Comment 9. Lines 72-73: Consider introducing the study species before the experiment as a way of justifying why those species were used.

Reply: We have move the text regarding study species earlier in the introduction before introduction of the experiment in line with the reviewer comment.

Comment 10. Lines 68-71: The authors might consider rephrasing or splitting this sentence into two.

Reply: We have divided the sentence into two.

Comment 11. Line 69: According to Table S1, the experiment ran from June 2017-October 2018, which would make it over two growing seasons and not over two full years. This is relevant, as winter dynamics could also be important for nutrient dynamics. Moreover, the winter season wouldn’t

count anyway because presumably all the earthworms died and had to be restocked during the next growing season.

Reply: In line with this comment, we now refer to the experiment as running over two growing seasons rather than over two years.

Comment 12. Line 80: The next section gives results for effects of earthworms on plant production after one and two growing seasons but it's not clear whether the analyses of plant N concentration, N sources, litter cover, PRS probes, PLFA only took place after the first season or both (line 80 refers to plant N and N sources after one growing season but it isn't mentioned for the second season or for the other analyses in this section).

Reply: We have specified throughout the Results section when measures were done after the first or second season.

Comment 13. Line 90: This sentence seems to be somewhat misleading. Of the four species-specific results described in this paragraph, there were significant effects of earthworms on N concentrations only in one of them (*Vac vit*), which was in the heath. There doesn't seem to be evidence that the increased plant N came from the litter layer in the heath, because that result was in *S. alpina*, a forb in the meadow. Moreover, the comment in lines 91-92 suggesting that there were relatively larger effects of an increase in N concentration in the heath should be rephrased if in fact the one heath plant was the only one to show significant changes in plant N (as worded, it sounds as though there were changes in the meadow but that the changes were larger in the heath). The heath-specific ¹⁵N results are also missing from this paragraph.

*Reply: We have now rephrased parts of this section to increase clarity. Note that more plant species responded to the earthworm treatment in the heath, which can be viewed as a stronger response, but we understand that the former wording could be misleading. In the revised version of the ms. we make clear that, in the heath, both *Festuca* and *Vaccinium vitis-idaea* have higher N concentration, and the effect is marginally significant for *V. myrtillus*. In the meadow, only *Festuca* had a higher N concentration, while there is no effect on *Saussurea* or *Bistorta*. Moreover, the uptake of ¹⁵N via earthworms is evident for *Festuca* in both habitats and *Saussurea* in the meadow, while no effect is found for the other species supporting that the increased N at least partly derive from litter.*

Comment 14. Lines 90-99: Were interactions with litter labeling tested for all species but only found to be significant for *S. alpina*? This result should be added for the heath species.

*Reply: This interaction was tested for all plants (Table S5) and this has been clarified in the revised version. Significant interactions effect for the labeled litter were found for *F. ovina* and *S. alpina**

Comment 15. Line 101: Typo - open parenthesis missing

Reply: Typo have been corrected

Comment 16. Line 103: Since it hasn't been mentioned before, it would be helpful to have an opening sentence here indicating that analyses of soil nutrients and microbial biomass were done using PRS and PLFA with very brief explanation of the acronyms.

Reply: We agree and have included an explanation of the acronyms

Comment 17. Line 106-107: Was the vertical mixing in both habitats?

Reply: Yes, and we clarified this in the revised ms.

Comment 18. Line 107: Likewise, it would be helpful to indicate here whether microbial biomass was measured from soil or litter samples.

Reply: We have specified that the measurements are from the upper 10 cm (covering both litter and the soil)

Comment 19. Line 109. Is the fungal biomass result a typo (meant to be $P > 0.3579$)?

Reply: Typo have been corrected

Comment 20. Line 139: Overall, the discussion is well-written and does a good job of placing the results in the context of the other drivers of change occurring in the Arctic.

Comment 21. Lines 211-212: Given that the authors did not record increases in mineral N in the soil solution, I'd suggest toning down this sentence to saying that the mobilization of N from litter and availability of N in earthworm casts were associated with higher plant N content rather than that they resulted in higher plant N.

Reply: We rephrased in line with the reviewer comment

Comment 22. Line 218: typo in this line

Reply: Typo have been corrected

Comment 23. Line 245: typo - no 's' in exists

Reply: Typo have been corrected

Comment 24. Line 249: Given that the experiment only lasted 2 years, the suggestion that the results show that earthworm effects could be long-lasting and potentially irreversible seems like an overreach.

Reply: The sentence has been re-phrased to state that the effects can be substantial, but without including a statement about the length and reversibility of the effect in line with the reviewer comment

REVIEWERS' COMMENTS:

Reviewer #3 (Remarks to the Author):

I approve of the changes made to the manuscript and think that the introduction in particular is greatly improved.

The manuscript will be a nice addition to the literature on the potential for organisms (in this case invasive organisms, which are not well studied in the North) to play an important role in ecosystem nutrient cycling.

Note: There remain a number of minor grammatical mistakes and other typos throughout the manuscript that need to be addressed prior to publication (e.g., lines 33, 39, 54, 67, 98, 167, 177).

From Jonatan Klaminder
Department of Ecology and Environmental Science
Umeå University
Jonatan.klaminder@umu.se

Point-by-point-response

Reviewer 3

Comment #1. I approve of the changes made to the manuscript and think that the introduction in particular is greatly improved. The manuscript will be a nice addition to the literature on the potential for organisms (in this case invasive organisms, which are not well studied in the North) to play an important role in ecosystem nutrient cycling. Note: There remain a number of minor grammatical mistakes and other typos throughout the manuscript that need to be addressed prior to publication (e.g., lines 33, 39, 54, 67, 98, 167, 177).

Reply: We have corrected the grammatical errors and typos identified by the reviewer. See track changes in main document.